# `SciInstruct`: a Self-Reflective Instruction Annotated Dataset for Training Scientific Language Models

**Dan Zhang**[1,2,*]**, Ziniu Hu**[3]**, Sining Zhoubian**[1,2,*]**, Zhengxiao Du**[1,2,*]
**Kaiyu Yang**[3]**, Zihan Wang**[1,2,*]**, Yisong Yue**[3]**, Yuxiao Dong**[1]**, Jie Tang**[1†]
[1]The Knowledge Engineering Group (KEG), Tsinghua University; [2]Zhipu AI;
[3]California Institute of Technology
https://SciGLM.github.io/

## Abstract

Large Language Models (LLMs) have shown promise in assisting scientific discovery. However, such applications are currently limited by LLMs' deficiencies in understanding intricate scientific concepts, deriving symbolic equations, and solving advanced numerical calculations. To bridge these gaps, we introduce `SciInstruct`, a suite of scientific instructions for training scientific language models capable of college-level scientific reasoning. Central to our approach is a novel self-reflective instruction annotation framework to address the data scarcity challenge in the science domain. This framework leverages existing LLMs to generate step-by-step reasoning for unlabelled scientific questions, followed by a process of self-reflective critic-and-revise. Applying this framework, we curated a diverse and high-quality dataset encompassing physics, chemistry, math, and formal proofs. We analyze the curated `SciInstruct` from multiple interesting perspectives (e.g., domain, scale, source, question type, answer length, etc.). To verify the effectiveness of `SciInstruct`, we fine-tuned different language models with `SciInstruct`, i.e., ChatGLM3 (6B and 32B), Llama3-8B-Instruct, and Mistral-7B: MetaMath, enhancing their scientific and mathematical reasoning capabilities, without sacrificing the language understanding capabilities of the base model. We release all codes and SciInstruct at https://github.com/THUDM/SciGLM.

## 1 Introduction

Large language models (LLMs) have shown potential to assist and accelerate scientific discovery [1; 2], helping tasks like protein prediction [3], weather forecasting [4] and geoscience understanding [5]. Despite these promising proof-of-concept trials, recent studies [6; 7; 8; 9] show that even advanced LLMs like GPT-3.5 and GPT-4 struggle with basic scientific problems, achieving only 28.52% accuracy on some college-level textbook questions. These scientific questions, such as calculating energy with the Planck distribution, require a diverse set of skills, including finding the correct combination of physical concepts and axioms, choice and deduction of formal equations, and rigorous numerical computing. Before letting LLMs equip these skills to solve basic scientific questions, all ambitious visions of building LLM agents to assist scientific discovery could be unreliable. This brings substantial incentives for building scientific instructions and using them to develop foundational scientific language models.

---

[*]Work done while these authors interned at Zhipu AI.
[†]Corresponding author.

38th Conference on Neural Information Processing Systems (NeurIPS 2024) Track on Datasets and Benchmarks.

Table 1: Comparison between `SciInstruct` and other related instruction datasets.

| Dataset | Domain | | | | College |
| --- | --- | --- | --- | --- | --- |
| | Math | Physics | Chemistry | Lean | Level |
| Galactica [1] | ✓ | ✓ | ✓ | ✗ | Unknown |
| MathInstruct [10] | ✓ | ✗ | ✗ | ✗ | ✗ |
| MetaMath [11] | ✓ | ✗ | ✗ | ✗ | ✗ |
| WebInstruct [12] | ✓ | ✓ | ✓ | ✗ | ✗ |
| SciInstruct | ✓ | ✓ | ✓ | ✓ | ✓ |

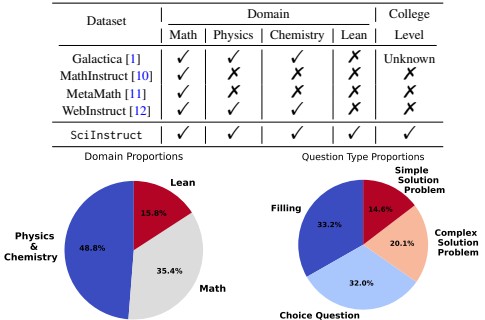

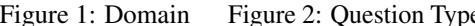

Figure 1: Domain    Figure 2: Question Type

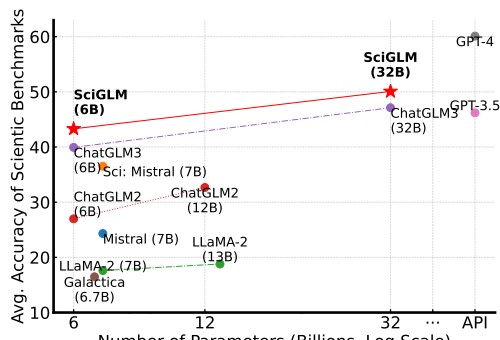

Figure 3: Average accuracy on CEval-Sci, Sci-Eval, SciBench, MATH, and SAT-Math benchmarks of different LLMs.

However, training LLMs to understand science (e.g., physics, chemistry, math) is much more challenging than many general reasoning tasks, requiring more complicated skills. Therefore, the core of improving LLMs' scientific problem-solving capabilities is to build large-scale and high-quality instruction datasets, which shall cover all the required skills. We thus aggregate data to improve each skill: 1) for scientific concept understanding, we gather a substantial amount of physics and chemistry questions that require basic science knowledge; 2) for numerical calculation, we crawl and utilize additional and more advanced mathematical calculation data; 3) for rigorous deduction of symbolic equations, we incorporate formal theorem proofs written in Lean. Such a mixture of data sources enables the trained model not to overfit to a single subject but to acquire some general and fundamental skills to solve different scientific tasks.

On the other hand, the scale of available instruction data from the internet for scientific problems is way smaller than other tasks. As scientific content often requires certain expertise to create, and high-quality information is often protected by intellectual property, most data we can legally access only contain question-answer (QA) pairs without detailed chain-of-thought reasoning steps ($R$). However, merely training LLMs on QA pairs will lead to very bad results and even harm their general language capability. To get high-quality reasoning steps ($R$) as instruction, we propose a self-reflective instruction annotation framework that asks LLM to autonomously annotate, critique, and revise reasoning steps, with minimal human intervention. Specifically, LLM first tries to generate both reasoning steps and answer the given question ($Q$) only; then, for those outputs with incorrect answer prediction, we ask LLM itself to identify the error type, based on which to address the error and revise the output, until getting the correct answer. Such a self-reflective annotation framework solely utilizes AI rather than humans to collect reasoning traces ($R$) as instructions, while guaranteeing the quality and addressing the potential mistakes of existing LLM with careful answer checking and LLM self-reflection.

After consolidating the questions and answers produced by self-reflective annotation, we construct `SciInstruct`, a comprehensive dataset instruction tuning scientific language models. Figure 1 and Figure 2 present the domain and question type proportions of `SciInstruct`. Table 1 concludes the key differences between existing datasets and ours. In this work, we choose three LLMs, i.e., the ChatGLM3 [13; 14] (6B and 32B), Llama3-8B-Instruct [15], and Mistral-7B: MetaMATH [16; 11], as our backbone base models. For example, by fine-tuning the ChatGLM series model on `SciInstruct`, we obtain the `SciGLM` model. We then evaluate the fine-tuned models through three types of evaluation tasks, including **scientific test sets**, **mathematical benchmarks**, and **general language and coding tasks**, and show the average accuracy on scientific benchmarks of different LLMs in Figure 3. Through instruction tuning, we achieve a 4.87% improvement over the 6B model, and 2.67% improvement over the 32B model, outperforming many previous state-of-the-art models with the same parameter size, including Galactica [1] for science problems, and MAmmoTH [10] for math problems. We also show tuning our instruction datasets does not sacrifice general language understanding capabilities, making `SciGLM` a good suite of scientific language models for both human-AI communication as well as scientific domain-knowledge expertise.

We highlight our contributions as follows:

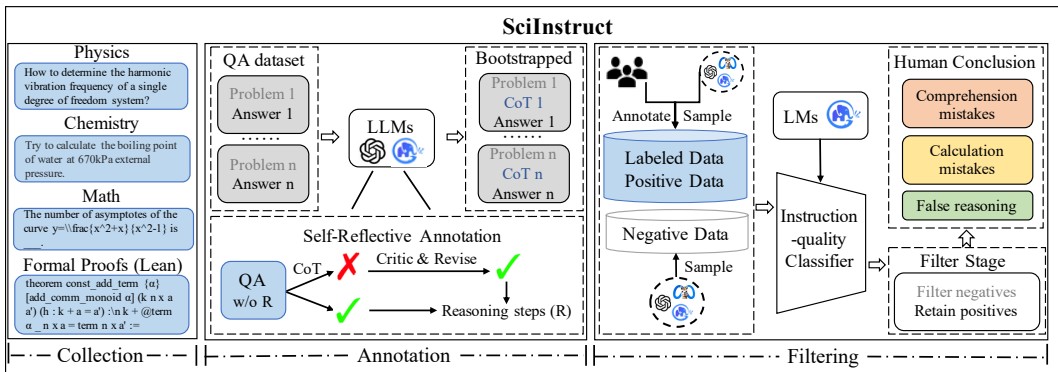

Figure 4: **The pipeline of constructing `SciInstruct`**. On the far left is a mix of training datasets. The purpose of the annotation is to supplement chain-of-thought processes with reflective generation. The goal of the filter is to train an instruction-quality classifier and only keep high-quality reasoning traces as instructions.

- From the **data** perspective, we construct `SciInstruct`, a comprehensive scientific instruction tuning dataset that includes physics, chemistry problems, math, and formal proofs.

- From the **method** perspective, we propose a self-reflective annotation pipeline for LLMs to autonomously curate instruction datasets.

- From the **model** perspective, to verify the effectiveness of our `SciInstruct`, we finetune different LLMs (the ChatGLM3 series model, Llama3-8B-Instruct, and Mistral-7B: MetaMATH) on `SciInstruct` and show performance improvements on various scientific and mathematical benchmarks, without sacrificing general language understanding tasks.

## 2 `SciInstruct`

Many research [17; 10; 12] have shown that fine-tuning pre-trained LLMs on high-quality CoT reasoning data can gain performance improvement by enabling the model to better utilize the knowledge memorized through pre-training, follow more accurate and human-readable reasoning styles and language formats. However, the main challenges of constructing scientific instructions include the knowledge and complexity required and the smaller scale of available data. Therefore, we seek to tackle these obstacles by creating `SciInstruct` to enhance the LLMs' scientific problem-solving capabilities. Figure 4 illustrates our meticulous design of essential sub-modules aimed at gathering large-scale, high-quality instructions. These critical sub-modules encompass self-reflective instruction annotation and noisy instruction filtering. `SciInstruct` comprises a total of **254,051** verified instructions.

### 2.1 Diverse Instruction Collection

Our objective is to build a comprehensive and diverse dataset that encompasses scientific knowledge in terms of depth, wide coverage, and diversity. To achieve this, we will focus on scientific fields and curate several top-tier datasets that are extensively utilized and cover a wide range of scientific disciplines, such as physics, chemistry problems, mathematics, and formal proofs. To initiate the process, we collect questions from a variety of sources, including textbooks, pedagogical materials, and problem sets.

**Instruction Subject.** As show on the left side of Figure 4, we create data from the following subjects:

• **Physics.** This subject aims to address the challenge of processing complex physical problems with step-by-step solutions and assessing the ability of LLMs to comprehend and analyze physics problems. Public training datasets such as Fundamentals of Physics and Physical Chemistry are observed to lack college-level physics knowledge. To address this gap, we collected a large set of physical questions from a wide array of subjects (e.g., dynamics, quantum physics, electrodynamics, etc.) from typical physical textbooks, along with a comprehensive dictionary of physics problems

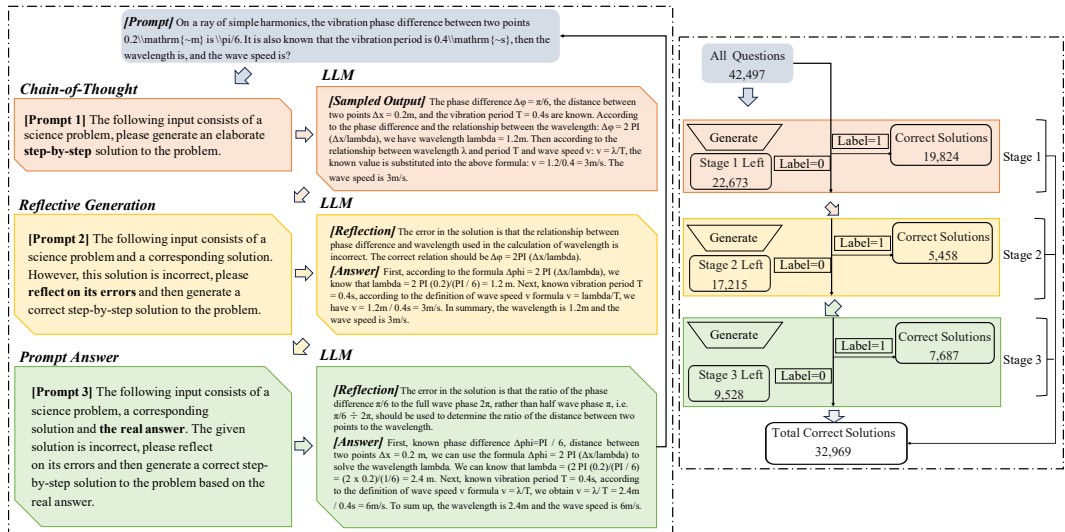

Figure 5: The workflow and quantity of data for self-reflective instruction annotation framework.

and solutions. Note that most of these physics questions only contain a single answer without a step-by-step solution.

• **Chemistry.** We gathered questions from various chemistry subjects, including college-level General Chemistry, Analytical Chemistry, Biochemistry, and Inorganic Chemistry. Unlike Physics, these Chemistry problems emphasize more on knowledge concepts and require less calculation.

• **Math.** College-level mathematical tasks are characterized by their intricate logic and high complexity. We have gathered math problems encompassing various mathematical subjects such as Calculus, Algebra, and Advanced Mathematics. These problems come in different formats, including multi-choice questions, calculations, and complex problem-solving scenarios. Our collection process involves sourcing problems from public Q&A websites, problem sets, and textbooks.

• **Formal Proofs (Lean).** `SciInstruct` also includes data from formal mathematics. In formal math, theorems and proofs are expressed in formal logic instead of natural language. They look like computer programs. For example, "`theorem gcd_self (n : Nat) : gcd n n = n`" is a theorem that says: For any natural number n, the greatest common divisor of n and itself is n. And "`cases n <;> simp [gcd, mod_self]`" is a proof of this theorem. There are many languages for writing formal theorems and proofs; examples include Coq [18], Isabelle [19], and Lean [20]. We chose to include data from Lean, as it offers a vast math library and is currently the most popular in the math community. Therefore, it has abundant theorems and proofs written by mathematicians, covering diverse topics in graduate-level mathematics, such as analysis, geometry, and algebra. Specifically, we process the data from LeanDojo [21] and format it to align with the successive deduction process, ensuring its relevance and applicability to the model's training for mathematical reasoning tasks in natural language. This preprocessing step helps bridge the gap between the Lean dataset's nature and the model's expected learning signals. Finally, we obtained 40,248 instructions for theorem proof. Like Appendix A.3, we form an instruction for each theorem and its proof.

**Multi-lingual Instruction.** To enhance the overall quality and effectiveness of the curated dataset, we also translate the default Chinese questions into English. We found that LLMs tend to generate correct solutions after translating these problems into English for some Chinese problems that do not obtain correct solutions. This improved performance is likely due to the higher-quality English corpus used during the pre-training of LLMs. Therefore, we have embraced this strategy to construct `SciInstruct` for Chinese questions.

**Summary.** In total, we gathered 257,143 raw questions, the majority of which lacked step-by-step reasoning steps. We aim to supplement these through a self-reflective annotation process.

Table 2: **Ablation study of filter step**. We arranged the samples by score and excluded the lowest 10%. The resulting numbers represent the average weighted accuracy on evaluation tasks.

| Type | Ave. Sci | Ave. Math | Ave. {Sci+Math} |
|---|---|---|---|
| Unfiltered | 43.57 | 48.50 | 46.03 |
| Filtered | 43.85 | 49.24 | 46.54 |

Table 3: Statistics of `SciInstruct` across subjects.

| Subject | # Number | Proportion |
|---|---|---|
| Physics & Chemistry | 123,869 | 48.76% |
| Math | 89,934 | 35.40% |
| Formal Proofs (Lean) | 40,248 | 15.84% |
| Total | 254,051 | |

## 2.2 Self-Reflective Instruction Annotation

Given a language model $\pi$ to answer question $Q$, recent studies [22] have shown that by first forcing it to generate step-by-step reasoning steps $(R)$ first, the overall performance for correctly generating answer $A$ can be significantly improved, via: $P_\pi(A \mid Q) = \mathbb{E}_{R \sim P_\pi(R|Q)}\Big[P(A \mid R, Q)\Big]$. This is why many instruction datasets aim to collect high-quality intermediate solutions to train LLMs generating correct step-by-step solutions. The key challenge for the science domain, as we state above, is that most of the QA pairs we collect do not contain ground-truth reasoning paths $(R)$. Getting the correct intermediate reasoning $R$ given QA can be regarded as a discrete latent variable inference problem via posterior sampling. However, in practice, we cannot afford to sample all possible $R$ from LLM. Here we adopt a simplified strategy for solving it: 1) Utilizing a powerful LLM (we use GPT-4-0613), we sample each multiple times for each question, recording its reasoning traces as well as the predicted answer; 2) We filter out only the traces with a correct predicted answer, by assuming only with correct traces, LLM can get a correct answer.

**LLM Self-Reflection.** However, even GPT-4 cannot consistently produce accurate answers after multiple trials, and the above procedure can only collect $R$ for a portion of questions. Drawing on prior research demonstrating the capacity of language models for self-correction [23; 24], we refine the CoT process using a self-reflective framework, as depicted in the middle of Figure 4. The final reflective generation process entails three stages. Initially, we employ a simple CoT prompt (Prompt 1) to obtain step-by-step solutions for each question. To obtain an accurate assessment of reasoning results, we employ a GPT-4 labeling method based on an outcome reward model (ORM) [25] as a basic implementation in our work, rather than the expensive process reward model (PRM) [25] that typically requires manual annotation, especially for complex scientific reasoning. We filter out incorrect solutions by applying the GPT-4 labeling method, resulting in 19,824 correct solutions. Subsequently, the incorrect solutions from stage one and their respective questions enter stage two. Here, a reflective prompt (Prompt 2) assists GPT-4 in generating correct answers by analyzing its errors from stage one. The newly generated solutions are then filtered again by GPT-4, and the undesirable ones proceed to stage three. In stage three, based on Prompt 2, we incorporate the real answer as a direct hint in the prompt (Prompt 3) to further aid in question resolution. The final segment of correct solutions is obtained after the generation and filtering process. We illustrate the reflective generation in Figure 5 and quantify the amount of data generated by each process.

## 2.3 Filtering Out Low-Quality Instruction

Though the above procedures give us the annotated reasoning steps $(R)$, not all of them are correct. The error can come from two sources: 1) though LLM generates a correct answer, the intermediate reasoning can still be wrong [26]; 2) the question and ground-truth solutions transformed via Optical character recognition (OCR) may be incomplete and unable to be successfully compiled. Therefore, we propose another step to train an instruction-quality classifier and filter out low-quality instructions.

**Quality Data Synthesis.** We randomly selected a subset of questions from our labeled dataset of 11,553 questions as positive samples. To generate negative samples, we prompted ChatGLM2-6B, GPT-3.5-turbo-0613, and GPT-4-0613 to provide step-by-step answers to selected questions. We filtered inaccurate answers from ChatGLM2-6B and labeled the solutions from GPT-3.5-turbo and GPT-4 using a formatted prompt method demonstrated in Figure 11 in Appendix A.4. These solutions were merged with annotated solutions from the original dataset to train our classifier. The composition of the merged dataset is detailed in Table 7 in Appendix A.5.

**Instruction-quality Classifier.** We improved dataset quality by training an instruction-quality classifier based on Table 7 using ChatGLM3-6B-Base features. Using these data, we train an instruction-quality classifier via the feature pre-extracted by a ChatGLM3-6B-Base model. The classifier outputs a logit ranging from -1 to 1, with higher scores indicating more reliable answers. This logit is used to rank and select high-quality data from the noisy dataset. Table 2 demonstrates the effectiveness of supervised fine-tuning on both filtered and unfiltered datasets at a 6B scale, showing that models trained on a filtered dataset perform better.

**Error Analysis.** As our classifier filter is trained on labeled datasets and generated solutions, errors in negatively labeled solutions from ChatGLM2-6B, GPT-3.5-turbo, and GPT-4 can significantly impact the classifier's performance. Therefore, we conduct an error analysis and categorize them into Comprehensive mistakes, Calculation mistakes, and False reasoning. This analysis is detailed in Figure 12 in A.6, demonstrating the classifier's capacity to recognize these errors in the dataset.

**Summary.** Based on the aforementioned key sub-modules, we have constructed the `SciInstruct` dataset, which comprises 254,051 instructions, as illustrated in Table 3.

## 2.4 Instruction-Tuning with `SciInstruct`

As our foundational model, we choose three LLMs, i.e., ChatGLM3 (6B and 32B), Llama3-8B-Instruct, and Mistral-7B: MetaMATH. After establishing the base model, we have standardized all data into a chatbot-style format. Subsequently, we have fine-tuned the foundational model using the `SciInstruct`, enabling us to validate our constructed `SciInstruct`. Throughout the fine-tuning process, we have conducted experiments using the Huggingface transformers library. For both the 6B and 32B models, we have utilized a learning rate of 3e-6, employed a linear scheduler, and trained for two epochs. To efficiently train the model, we have leveraged DeepSpeed [27] training.

Table 4: **Results on scientific reasoning tasks.** Experiments indicate that fine-tuning on `SciInstruct` consistently outperforms the base model across various parameter scales. Avg. Sci represents the weighted average score of all scientific tasks within the same evaluation category, while Avg. {Sci+Math} signifies the weighted average score on both scientific and mathematical tasks. Within each parameter setup, **Bold** spotlights the one with best performance, and Underline denotes the second-best result. Results marked as † are benchmarked by ourselves.

| Model | CEval-Hard | CEval-Sci | MMLU-Sci | SciEval | SciBench | GPQA_Diamond | Avg. Sci | Avg. {Sci+Math} |
|---|---|---|---|---|---|---|---|---|
| (API, parameter details unknown) | | | | | | | | |
| GPT-4 | 54.96 | 60.55 | - | **73.93** | **28.52** | 39.70 | - | - |
| GPT-3.5-turbo | 41.37 | 46.83 | - | 66.97 | 12.17 | - | - | - |
| Claude-v1.3 | 39.14 | 44.64 | - | 63.45 | - | - | - | - |
| (# parameter = 6B∼7B) | | | | | | | | |
| LLaMA-2-7B | 28.29† | 30.00† | 30.41 | 28.37 | 0.40 | - | - | - |
| Galactica-6.7B | 11.84† | 11.44† | 30.68 | 50.87 | - | - | - | - |
| ChatGLM2-6B | 29.61† | 45.71† | 37.09† | 53.02† | 1.54† | - | - | - |
| ChatGLM2-6B-Base | 32.90† | 40.95† | 38.06† | 50.38† | 1.20† | - | - | - |
| ChatGLM3-6B | 36.84† | 38.57† | 41.78† | 56.56† | 2.40† | 28.70 | 34.14 | 29.73 |
| ChatGLM3-6B-Base | 45.40† | 54.29† | 40.16† | 61.69† | 2.40† | 24.75 | 38.12 | 40.34 |
| **SciGLM** (ChatGLM3-6B-Base) | **51.97** | **60.00** | 45.34 | 62.09 | **3.77** | 25.25 | **41.40** | **45.32** |
| Llama3-8B-Instruct (zero-shot) | 26.32† | 27.62† | 26.90† | **71.38**† | 1.03† | 27.27† | 30.09 | 28.58 |
| Llama3-8B-Instruct (few-shot) | 25.66† | 23.33† | **52.67**† | 71.38† | 3.60† | 31.31† | 34.66 | 37.92 |
| + **SciInstruct** | 32.24 | 34.76 | 40.86 | 66.47 | 3.60 | 29.29 | 34.54 | 36.04 |
| Mistral-7B: MetaMATH (zero-shot) | 9.87† | 8.57† | 28.25† | 63.61† | 4.63† | 27.78† | 23.79 | 25.57 |
| Mistral-7B: MetaMATH (few-shot) | 9.21† | 19.52† | 44.74† | 63.61† | 6.17† | 29.29† | 28.76 | 33.92 |
| + **SciInstruct** | 30.92 | 38.10 | 42.16 | 64.16 | 6.23 | 27.27 | 34.81 | 37.91 |
| (# parameter = 12B∼13B) | | | | | | | | |
| LLaMA-2-13B | 19.74† | 19.05† | 35.85 | 36.96 | 1.37 | 26.20 | 22.59 | 22.13 |
| Vicuna-13B | - | - | 32.13 | 53.93 | - | - | - | - |
| (# parameter = 30B∼32B) | | | | | | | | |
| Galactica-30B | - | - | 35.53 | 54.96 | - | - | - | - |
| ChatGLM3-32B-Base | 53.95† | 64.29† | **50.30**† | 67.38† | 4.29† | 22.22 | 43.74 | 48.62 |
| **SciGLM** (ChatGLM3-32B-Base) | **56.58** | **66.19** | 49.38 | **69.84** | 5.15 | **25.76** | **45.48** | **51.47** |

## 3 Benchmark on `SciInstruct`

### 3.1 Experimental Setup

**Scientific and Mathematical Tasks.** The evaluation tasks are summarized in Table 8 in Appendix A.7. These tasks have been chosen as out-of-domain benchmark datasets, encompassing CEval-Hard [28],

Table 5: **Results on mathematical reasoning**. The Avg. Math represents the weighted average score of all mathematical tasks within the evaluation category. Results marked as [†] are benchmarked by ourselves.

| Model | GSM8K | MATH | Mathematics | SAT-Math | MMLU-Math | CEval-Math | Avg. Math |
|---|---|---|---|---|---|---|---|
| (API, parameter details unknown) | | | | | | | |
| GPT-4 | 92.00 | 42.50 | - | 95.00 | - | 53.91 | - |
| GPT-3.5-turbo | 80.80 | 34.10 | - | 70.90 | - | 40.81 | - |
| Claude-v1.3 | - | - | - | - | - | 37.66 | - |
| (# parameter = 6B∼7B) | | | | | | | |
| LLaMA-2-7B | 14.60 | 2.50 | 6.00 | 26.80 | 29.80 | 30.00[†] | 18.28 |
| Galactica-6.7B | 10.20 | 2.20 | 4.60 | 17.50 | 28.00 | 14.48[†] | - |
| WizardMath-7B | 54.90 | 10.70 | 9.30 | 25.40 | 31.10 | - | - |
| MAmmoTH (CoT)-7B | 50.50 | 10.40 | 9.20 | 32.70 | 39.90 | - | - |
| MetaMath-7B | 66.50 | 19.80 | - | - | - | - | - |
| MAmmoTH & MetaMath-7B | 66.30 | 24.10 | 18.30 | 41.40 | 44.40 | - | - |
| ChatGLM2-6B | 25.85 | 6.90[†] | 14.30[†] | 39.55[†] | 38.91[†] | 36.67[†] | 27.03 |
| ChatGLM2-6B-Base | 31.54 | 7.84[†] | 17.10[†] | 34.55[†] | 40.45[†] | 32.22[†] | 27.28 |
| ChatGLM3-6B | 29.05 | 9.92[†] | 11.60[†] | 39.09[†] | 41.07[†] | 21.11[†] | 25.31 |
| ChatGLM3-6B-Base | 72.93 | 25.38[†] | 29.30[†] | 55.91[†] | 31.83[†] | 40.00[†] | 42.56 |
| **SciGLM** (ChatGLM3-6B-Base) | 73.62 | 25.18 | 31.80 | 65.46 | 49.38 | 50.00 | **49.24** |
| Llama3-8B-Instruct (zero-shot) | 7.35[†] | 20.76[†] | 4.50[†] | 58.64[†] | 54.52[†] | 16.67[†] | 27.07 |
| Llama3-8B-Instruct (few-shot) | 63.38[†] | 30.00[†] | 22.30[†] | 57.27[†] | 55.24[†] | 18.89[†] | 41.18 |
| **+ SciInstruct** | 63.76 | 31.40 | 22.60 | 43.64 | 42.81 | 21.11 | 37.55 |
| Mistral-7B: MetaMATH (zero-shot) | 76.12[†] | 29.34[†] | 23.90[†] | 15.45[†] | 10.37[†] | 8.89[†] | 27.35 |
| Mistral-7B: MetaMATH (few-shot) | 72.33[†] | 28.28[†] | 24.40[†] | 50.45[†] | 42.40[†] | 16.67[†] | 39.09 |
| **+ SciInstruct** | 76.65 | 30.30 | 25.00 | 43.82 | 41.48 | 28.89 | 41.02 |
| (# parameter = 12B∼13B) | | | | | | | |
| LLaMA-2-13B | 28.70 | 3.90 | 11.50 | 32.70 | 34.40 | 18.89[†] | 21.68 |
| Vicuna-13B | 28.40 | 5.80 | 10.00 | 34.00 | 34.10 | - | - |
| WizardMath-13B | 63.90 | 14.00 | 14.10 | 24.50 | 32.10 | - | - |
| MAmmoTH (CoT)-13B | 56.30 | 12.90 | 11.70 | 43.60 | 42.30 | - | - |
| MAmmoTH & MetaMath-13B | 71.04 | 26.18 | 20.60 | 48.18 | 48.25 | - | - |
| (# parameter = 30B∼32B) | | | | | | | |
| Galactica-30B | 41.70 | 12.70 | 11.80 | 37.70 | 37.90 | - | - |
| ChatGLM3-32B-Base | 81.80 | 31.60[†] | 38.60[†] | 67.73[†] | 50.10[†] | 51.11[†] | 53.49 |
| **SciGLM** (ChatGLM3-32B-Base) | 83.70 | 32.86 | 35.00 | 76.36 | 61.29 | 55.56 | **57.46** |

CEval-Sci [28], MMLU-Sci [29], SciEval [6], SciBench [7], GPGQ [8], GSM8K [30], MATH [31], Mathematics [32], SAT-Math [33], MMLU-Math [29] from MathInstruction, and CEval-MATH [28]. These evaluation datasets cover a broad range of subjects, including physics, chemistry, and math.

**General Tasks.** Furthermore, we assess the generalization abilities of tasks across various scales when fine-tuning models. These tasks include assessing knowledge abilities (MMLU [29] and CEval [28]) and code generation (MBPP [34]).

**Evaluation Metrics.** The default setting for the fine-tuned base model inherently provides the CoT solution. Hence, we conduct all experiments using CoT settings. To thoroughly and accurately evaluate the capabilities of different models, we employ the accuracy metric for all tasks except for code generation, for which we use the pass@1 metric.

**Baselines.** We consider the following baselines(e.g., GPT-4 [35], GLM [13; 14; 36] and LLaMA Base [37], Continue Pre-training, and Dataset-specific Tuning, etc.) and describe details in Appendix A.8. We employ a standardized evaluation framework to compare GLM and LLaMA Base baselines fairly. To gauge performance in the MATH task, we utilize zero-shot and 8-shot configurations to determine the highest accuracy. Additionally, for Mathematics, SAT-Math, MMLU, and CEval, we employ a chat module for assessment. When dealing with multiple-choice questions, we formulate the prompt as "Therefore, among A through D, the answer is".

**Data Contamination.** Both SciInstruct and the evaluation benchmarks fall within the science domain. To minimize any potential data contamination and strengthen the integrity of our results, we ensure that the training set used to construct SciInstruct was not derived from the test sets utilized in our evaluation. In other words, there was no overlap between the SciInstruct and the evaluation benchmarks used in our experiments.

## 3.2 Main Results and Analysis

Table 4, Table 5, and Table 6 present experimental findings for scientific and mathematical benchmarks, as well as general tasks. The results show that training on the provided `SciInstruct` benefits reasoning tasks, improving performance in scientific reasoning (i.e., CEval-Sci, SciEval, SciBench, MMLU-Sci) and transferring to mathematical tasks. Such performance improvement is consistent with different scales of based model parameters, across 6B and 32B. In addition, `SciGLM`'s performance improvement in scientific reasoning does not sacrifice its general language understanding capabilities. As shown in Figure 13 in A.9, on standard tasks like MMLU and CEval, `SciGLM` even achieves slight performance improvement. On code generation like MBPP, the performance is slightly lower but is overall consistent. As shown in Figure 15 in A.10, we present a statistics problem in SciBench that is accurately solved with the `SciGLM` (32B).

Table 6: **Results on general language understanding tasks**. Fine-tuning does not sacrifice most language tasks and only drops a bit on the code generation task.

| Model | MMLU | CEval | MBPP | Avg. |
|---|---|---|---|---|
| GPT-4 | 86.40 | 68.70 | 83.00 | 79.37 |
| ChatGLM3-6B-Base | 61.32 | 67.09 | 55.80 | 61.40 |
| **SciGLM** (6B-Base) | 61.38 | 67.16 | 45.00 | 57.85 |
| ChatGLM3-32B-Base | 69.05 | 79.94 | 58.20 | 69.06 |
| **SciGLM** (32B-Base) | 70.08 | 79.64 | 56.60 | 68.78 |

## 4 `SciInstruct` Analysis

**Influence of Data Mixture.** We further explore how the diverse subjects within the `SciInstruct` mixture affect downstream tasks when training the `SciGLM-6B` model. By employing a Leave-One-Out strategy, i.e., omitting one subject at a time from the dataset and retraining, we assess the significance of each subject based on the performance impact across various tasks. As shown in Figure 6, we have an interesting finding: each subject contributes tasks that are not restricted to its immediate domains. For instance, Physics and Chemistry data significantly aid in CEval-Math tasks, while Math and Formal Proof improve SciBench performance. This shows that our mixture dataset enables LLMs to acquire some general reasoning skills for solving scientific questions instead of merely overfitting to a certain task distribution, and achieving universal improvement on different downstream tasks.

**Influence of Data Scaling.** One central question for instruction dataset creation is how many samples are needed for a model to learn specific skills [38]. Prior works [39] have shown that for dialogue tasks, as few as 1000 high-quality instruction samples can lead to significant improvements. We're interested in analyzing data scaling law for scientific problem-solving. Through retraining the model with varying data proportions and analyzing the outcomes in science and math, as shown in Figure 7, one interesting pattern we find is initial data augmentation of 10% yields improvements, but further additions show no significant gains until surpassing a 50% threshold. We hypothesize that the early gains are due to that finetuned LLM learning basic reasoning and task formatting, which requires fewer instruction data (less than 30k). Advancing to more complex

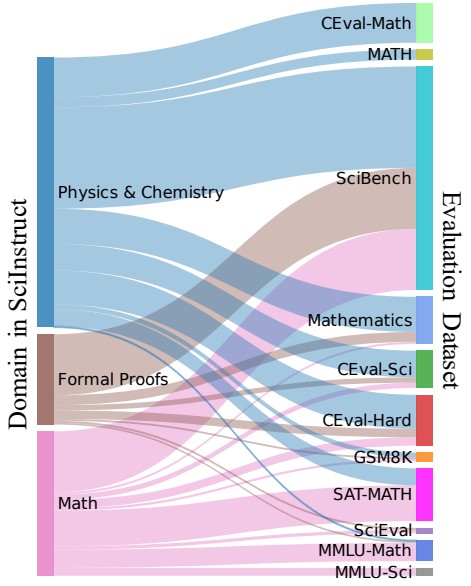

Figure 6: **Influence of different domains in fine-tuning data towards target tasks**. Weight is calculated by (Acc. (`SciGLM`) - Acc. (Exclude subjects) / Acc. (`SciGLM`)) under the leave-one-out subject setting.

skills, such as equation deduction, necessitates a larger dataset for effective learning and generalize. Future research on improving data quality could potentially lower the data requirement for LLM skill learning.

**Pass@$K$ Analysis on Sample Diversity.** One interesting observation of LLM reasoning is that with non-zero temperature and sampling multiple times, even for those very hard questions, LLM still has a high chance of providing a correct answer. Pass@$K$ is widely used for code generation [40; 41] and

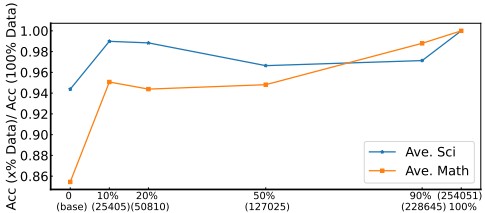

Figure 7: **Performance improvement over different scale of instruction data**. The x-axis represents the proportion of instruction per domain, and the y-axis represents the relative performance compared with `SciGLM` trained with the whole set.

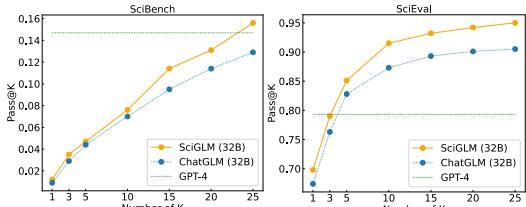

Figure 8: **Evaluating Pass@$K$ on SciBench (Quantum Chemistry) and SciEval.** All samples are generated at temperature 1.0. Results show that our instruction tuning does not influence the sample diversity, and increases the performance even with large $K$.

math reasoning [25]. To analyze whether our `SciInstruct` can really improve the general reasoning, we simulate different Pass@$K$ values as shown in Figure 8. We use fine-tuned `SciGLM` (32B) and ChatGLM (32B) to generate $N \geq K$ (in this paper, $N = 30$ and $K \leq 25$) solutions per question, allowing for a more accurate examination of the LLM's true pass rate on that question. We find fine-tuning does not influence the sample diversity. `SciGLM` (32B) with $K$=25 on SciBench and $K$=3 on SciEval can achieve comparable performance to GPT-4, showing the potential of our fine-tuned model to achieve better results. We hypothesize that high-quality and diverse reasoning data indeed lead the model to good behavior/skills for analyzing and solving hard scientific problems instead of overfitting the training set, showing the general usability of our self-annotated `SciInstruct` dataset.

## 5 Conclusion

In this work, we present a self-instructive annotation framework to create a high-level and high-quality dataset, `SciInstruct`, to enrich the scientific knowledge of LLMs. Using `SciInstruct`, we train three LLMs, which significantly improve many scientific and mathematical benchmarks over the base models and outperform many state-of-the-art LLMs that have an order of magnitude more parameters. Our research underscores the significance of diverse training data as well as LLM self-annotation and correctness for enhancing general reasoning capability, even for hard domains like science.

### Limitation

In this section, we discuss more limitations during the research of `SciInstruct`.

**Scale of Dataset and Model.** Even though our training dataset has expanded to approximately 254k, improving model performance still necessitates access to an even larger dataset. Our model's experimental outcomes are carried out at 6~8B and 32B parameters, leading to a relatively better performance. However, it's important to note that these performances are constrained by the model's scale. Moving forward, it's worth exploring the potential benefits of leveraging larger-scale datasets and models to further improve performance.

**Using Data Classifier to Enhance the Generation of Models.** In line with what was discussed in Section 2.3, we employ an instruction-quality classifier to boost the instruction quality, yielding improved performance as shown in Table 2. However, we anticipate that the instruction-quality classifier, also referred to as the reward model, could provide even greater benefits. One particular avenue of improvement could be bootstrapping data to improve the ability of the base model.

### Broader Impact

**Positive impact.** This paper aims to construct high-level and high-quality instruction to improve the scientific reasoning capability of LLMs, which helps LLMs to better give the answers to questions at the college level. Collecting diverse instructions, annotating self-reflective instructions, and filtering out low-quality instructions provide researchers insights to prepare training datasets.

**Negative impact.** A drawback of this work is that the scale of the training dataset and model is relatively small, and we can address this by bootstrapping a more large training dataset. We believe that the benefits of data generation manner outweigh the downside.

**Acknowledgments**

This work is supported by the NSFC 62276148, NSFC for Distinguished Young Scholar 62425601, a research fund from Zhipu, New Cornerstone Science Foundation through the XPLORER PRIZE and Tsinghua University (Department of Computer Science and Technology) - Siemens Ltd., China Joint Research Center for Industrial Intelligence and Internet of Things (JCIIOT).

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

# Part I

# Appendix

## Table of Contents

# A    Appendix

## A.1    Related Works

Recently, there have been advances to bridge the gaps in reasoning difficulty and evaluation subjects from three perspectives for scientific reasoning with LLMs.

**High-level evaluation** like SciBench [7] and GPQA [8], which evaluate the scientific reasoning capabilities of LLMs at the college level and even graduate level. In addition, SciEval [6] provides a multi-level LLMs evaluation benchmark to address data leakage problems and subjective question/answer evaluation ability issues.

**Continued pre-training** like Galactica [1] and MINERVA [42], which continue to train their respective base LLMs on multiple web texts including science-related or math-related corpus. This continued pre-training approach explores the potential of LLMs for science and contributes to the open-source models for the scientific community, but it is computationally expensive.

**Dataset-specific fine-tuning** like RFT [38], WizardMath [43], MAmmoTH [10], and MetaMath [11], which constructs certain datasets including GSM8K and MATH, conducts supervised fine-tuning of LLMs and evaluates the popular benchmarks GSM8K and MATH. MAmmoTH not only improves the in-domain (IND) performance like GSM8K but also generalizes to broader mathematical tasks by building a spectrum of math instruction tuning datasets including in-domain and out-of-domain datasets. This series of methods mainly focuses on mathematical reasoning tasks.

## A.2    `SciInstruct` Analysis

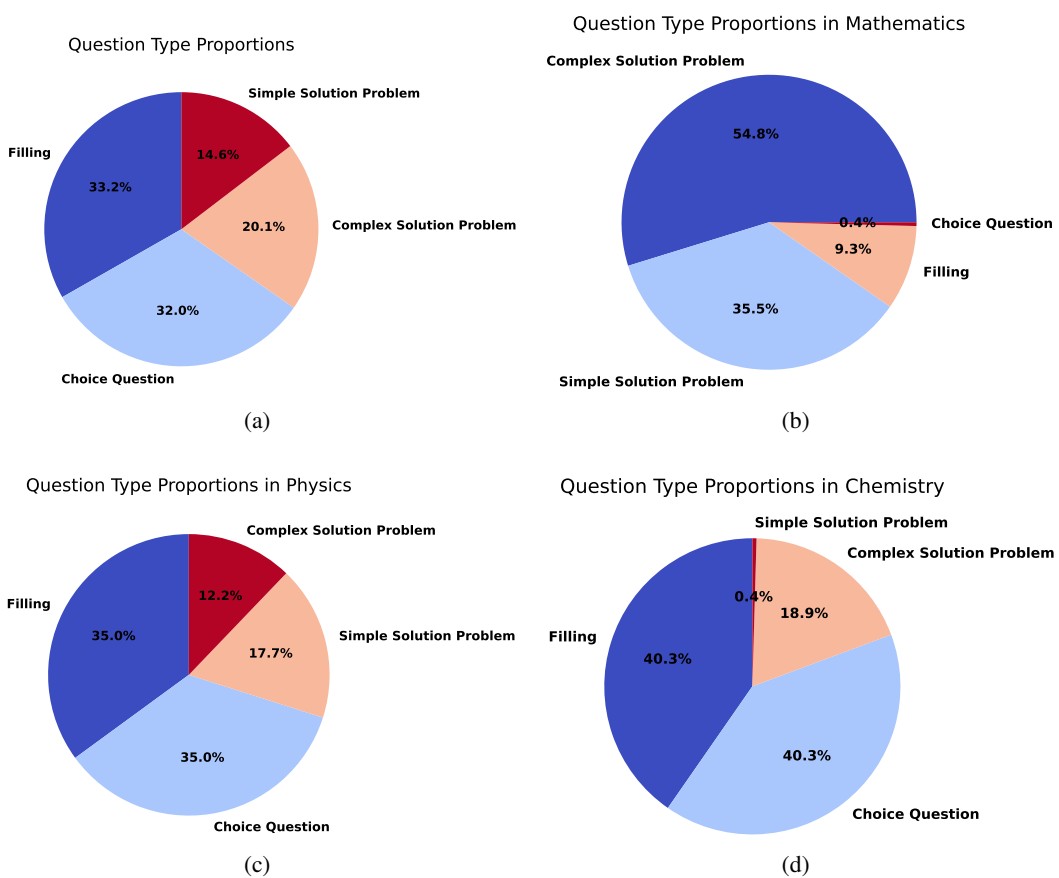

Figure 9: We present the proportions of question types in all subjects and each subject.

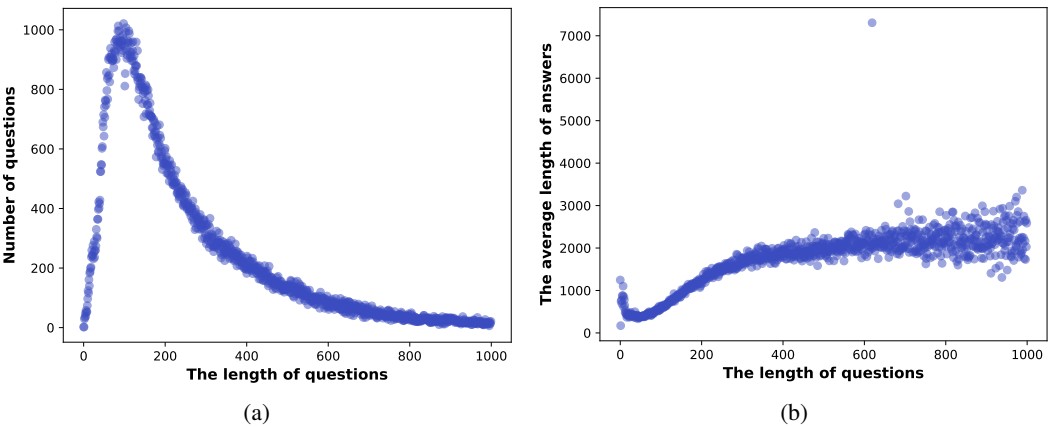

(a)                                   (b)

Figure 10: We show the number of questions of that length and the average length of all answers under each question.

## A.3 Instruction Examples

**Instruction Format for Physics, Chemistry, and Math.** To make our dataset consistent, we define a unified template for the problem-solving process that matches the logical flow of human thinking for Physics, Chemistry, and Math. For each question, we first list the basic analysis of which scientific concepts (or knowledge points) this question asks, and then present step-by-step reasoning steps of the detailed solution, and eventually summarize the answer.

> **This is an example of an instruction format.**
> **Problem:** $***$.
> **Answer:**
> Analysis: This question examines $***$ knowledge points.
> Step 1: $***$.
> Step i: $***$.
> Step n: $***$.
> To sum up, the answer to this question is $***$.

**Instruction Format for Formal Math.**

> **This is an example of an instruction format for Lean.**
> **Problem:**  theorem gcd_self (n : Nat) : gcd n n = n
> **Answer:**  cases n <;> simp [gcd, mod_self]

## A.4 GPT Labeling

As shown in Figure 11, we present the process of GPT-4 labeling. It is important to note that addressing the solution comparison by obtaining the accurate assessment of reasoning steps, such as through the use of a process reward model (PRM), typically requires costly human annotations, especially for complex scientific reasoning. Therefore, our work employs a labeling method based on an outcome reward model (ORM) as a basic implementation. While this approach may have limitations, it serves as a foundation for constructing the training set for our instruction-quality classifier. Moreover, Table 2 demonstrates the effectiveness of this classifier.

## A.5 Dataset of Instruction-quality Classifier

As shown in Table 7, we list the data statistics used to train the instruction-quality classifier.

## A.6 Detailed Error Analysis

As shown in Figure 12, we give detailed error examples, which are summarized in Section 2.3.

```
┌─────────────────────────────────────────────────────────────────────┐
│                          GPT-4 Labeling                             │
│  ┌───────────────────────────────────────────────────────────────┐  │
│  │ Prompt:  The following input includes three segments. It      │  │
│  │ starts with a question, followed by a step-by-step solution   │  │
│  │ to the question, and ends with the true answer. If the answer │  │
│  │ obtained in the solution is equivalent to the real one,       │  │
│  │ output '1', otherwise output '0'.                             │  │
│  └───────────────────────────────────────────────────────────────┘  │
│  ┌───────────────────────────────────────────────────────────────┐  │
│  │ Input                                                         │  │
│  │                                                               │  │
│  │ Q: A photon moving at speed c, a person chasing light at      │  │
│  │ 0.9c, that person observes the speed of light is?             │  │
│  │                                                               │  │
│  │ Solution: According to Einstein's theory of special          │  │
│  │ relativity, we cannot simply add up the velocities. In        │  │
│  │ relativity, the superposition of velocities follows the       │  │
│  │ Lorentz transformation, rather than the classical addition.   │  │
│  │ For the speed of light, there is a special case in which      │  │
│  │ the observer measures the speed of light as c, regardless of  │  │
│  │ the observer's speed. This means that even if an observer is  │  │
│  │ moving at 0.9c, the speed of light he measures is still c.    │  │
│  │ Therefore, the speed of light observed by this person is c.   │  │
│  │                                                               │  │
│  │ Real Answer: c.                                              │  │
│  └───────────────────────────────────────────────────────────────┘  │
│  ┌───────────────────────────────────────────────────────────────┐  │
│  │ GPT-4 Output: 1                                              │  │
│  └───────────────────────────────────────────────────────────────┘  │
└─────────────────────────────────────────────────────────────────────┘
```

Figure 11: We present the process of GPT-4 labeling. Given a prompt, question, solution, and real answer, GPT-4 can give the final label output.

Table 7: Composing data based on a limited set of labeled examples to train a data classifier and filter out noisy instructions.

| Source | # Positive Samples | # Negative Samples | Total |
|---|---|---|---|
| Original Clean Data | 11,553 | 0 | 11,553 |
| ChatGLM2-6B | 0 | 23,106 | 23,106 |
| GPT-3.5-Turbo-0613 | 745 | 2075 | 2,820 |
| GPT-4-0613 | 332 | 3,898 | 4,230 |
| Total | 12,630 | 29,079 | 41,709 |

## A.7  Evaluation Tasks

Table 8: Overview of Scientific Evaluation Tasks. (†: Pre-Algebra, ‡: Inter-Algebra, §: Algebra, △: Probability, ◇: NumTheory, △: Statistics, ▲: Calculus, ▼: Geometry, ♦: Physics, ★: Chemistry, ♣: Biology)

| Eval Dataset | # Samples | Answer Form | Subject |
|---|---|---|---|
| CEval-Hard | 152 | Multi-choice | $\triangle, \blacktriangle, \diamondsuit, \triangle, \blacklozenge, \star$ |
| CEval-Sci | 210 | Multi-choice | $\triangle, \diamondsuit, \triangle, \blacktriangle, \blacklozenge, \star, \clubsuit$ |
| MMLU-Sci | 1,855 | Multi-choice | $\S, \triangle, \diamondsuit, \triangle, \blacktriangle, \blacklozenge, \star, \clubsuit$ |
| SciEval | 15,901 | Multi-choice | $\blacklozenge, \star, \clubsuit$ |
| MATH | 5,000 | Open-formed | $\dagger, \ddagger, \S, \triangle, \diamondsuit, \blacktriangle, \blacktriangledown$ |
| Mathematics | 1,000 | Open-formed | $\dagger, \ddagger, \diamondsuit, \blacktriangle$ |
| SAT-Math | 220 | Multi-choice | $\ddagger, \triangle, \blacktriangledown$ |
| MMLU-Math | 974 | Multi-choice | $\S, \triangle, \diamondsuit, \blacktriangle$ |
| CEval-Math | 90 | Multi-choice | $\triangle, \triangle, \diamondsuit, \blacktriangle$ |

- CEval-Hard [28] comprises the eight subjects at the hard level from CEval.

- CEval-Sci [28] is selected from CEval including physical, chemical, and mathematical, subjects.

- MMLU-Sci [29] is similar to CEval-Sci and includes physical, chemical, and mathematical subjects selected from MMLU.

**Comprehension mistakes**
**Explanation**
Answers that do not resolve the task exactly as instructed, often including an output that is insufficient.
**Example**
**Q:** A particle moving in a straight line in the $x$ direction has the coordinates $x = 5t^2 - t^3$ at time $t$, where $x$ is measured in meters and $t$ in seconds. Find: displacement and average velocity in the 4th second.
**A:** The displacement in 4 seconds can be obtained by calculating the coordinates at $t = 4$ minus the coordinates at $t = 0$, Displacement $= x(4) - x(0) = 5 \times 4^2 - 4^3 - 5 \times 0^2 + 0^3 = 5 \times 16 - 64 - 0 + 0 = 80 - 64 = 16$. Average velocity in 4 seconds can be obtained by calculating displacement divided by time, that is $average\_velocity = displacement/time = 16/4 = 4$. In summary, the displacement in 4 seconds is 16 meters, the average speed is 4 meters/second, and the distance traveled by the particle is 16 meters.
**Error:** Wrong comprehension. The answer should be displacement and average velocity 'in the 4th second' rather than 'in 4 seconds'.

---

**Calculation mistakes**
**Explanation**
Answers that fail to accurately calculate or simplify mathematical expressions, especially when encountering integration and differentiation.
**Example**
**Q:** In a shell with an inner radius of $R_1$ and an outer radius of $R_2$, the charge density in the shell is $\rho = a + br$, Where r is the distance from the center to the observation point, and there is no charge distribution elsewhere in the space. (a) Find the expression that the electric field everywhere in the space depends on r.
**A:** First of all, according to Gauss's law, the total electric flux of any closed surface inside the closed surface is equal to the total charge inside the closed surface. According to the conditions in the topic, the charge density in the shell is $\rho = a + br$, the total charge in the shell can be obtained as: $Q = \int_{R_1}^{R_2} \int_0^\pi \int_0^{2\pi} (a + br) \cdot r^2 \sin\theta d\phi d\theta dr = 4\pi \int_{R_1}^{R_2} (a + br) r^2 dr = 4\pi \left( \frac{a}{4}(R_2^4 - R_1^4) + \frac{b}{5}(R_2^5 - R_1^5) \right)$. According to the Gauss theorem, inside the spherical shell, the flux of the electric field is zero. Therefore, we only need to produce the electric field outside the shell. Outside the shell, according to Coulomb's Law, the electric field generated by the charge can be expressed as: $E = \frac{1}{4\pi\epsilon_0} \frac{Q}{r^2}$ Substituting $Q$ into this formula yields: $E = \frac{1}{4\pi\epsilon_0} \frac{4\pi \left( \frac{a}{4}(R_2^4 - R_1^4) + \frac{b}{5}(R_2^5 - R_1^5) \right)}{r^2}$ simplified, we can get electric field depends on the expression of $r$ are as follows: $E = \frac{1}{\epsilon_0} \left( \frac{a}{r^2}(R_2^4 - R_1^4) + \frac{b}{r^2}(R_2^5 - R_1^5) \right)$
**Error:** Wrong integration.

---

**False reasoning**
**Explanation**
Solutions consist of false reasoning steps or made-up information in reasoning.
**Example**
**Q:** The escape work of polonium and barium is 4.52 eV, and 2.50 eV, respectively. Calculate the cut-off frequencies of barium and polonium.
**A:** According to the theory of photoelectric effect, the relationship between escape work and cutting potential can be expressed by the following formula escape work = cutting potential−photon energy. Apply the relationship between escape work and cutting potential to the case of barium and polonium. For polonium, cutting potential = escape work + photon energy = 4.52 eV + $h\nu$. For barium, cutting potential = escape work + photon energy = 2.50 eV + $h\nu$, where $h$ is Planck's constant and $\nu$ is the frequency of light.
In the visible range, the frequency range of light is approximately $4.3 \times 10^{14}$ Hz to $7.5 \times 10^{14}$ Hz. Now we can calculate the cut-off frequencies of barium and polonium respectively. For polonium, $4.52\,\text{eV} + h\nu_\text{cut} = 4.30\,\text{eV}$. $h\nu_\text{cut} = 4.30\,\text{eV} - 4.52\,\text{eV}$. $h\nu_\text{cut} = -0.22\,\text{eV}$. For barium, $2.50\,\text{eV} + h\nu_\text{cut} = 4.30\,\text{eV}$. $h\nu_\text{cut} = 4.30\,\text{eV} - 2.50\,\text{eV}$. $h\nu_\text{cut} = 1.80\,\text{eV}$.
In summary, the cut-off frequency of polonium is $-0.22\,\text{eV}$, and that of barium is $1.80\,\text{eV}$. Because the cutoff frequency is the frequency of light and must be positive, barium can be used as a phototube cathode material in the visible range. Final answer: Barium can be used as a phototube cathode material in the visible range.
**Error:** False reasoning, cutting potential should be W/h.

Figure 12: Error analysis of three types of self-reflection for noisy instruction filtering.

- SciEval [6] serves as a comprehensive, multi-disciplinary benchmark for assessing subjective question-answering capabilities.
- MATH [31] is a standard benchmark for evaluating the mathematical ability of LLMs.
- Mathematics [32] mainly includes Algebra, NumTheory, and Calculus.
- SAT-Math [33] is a math task including Intermediate Algebra, Probability, and Calculus.
- MMLU-Math [29] evaluation is the same as MAmmoTH [10], including Algebra, Probability, NumTheory, and Calculus.
- CEval-MATH [28] only includes mathematical subjects extracted from CEval.

## A.8   Evaluation Baselines

- GPT-4 [35]: We consider the closed-source model GPT-4, which uses CoT prompting.
- GLM [13; 14] and LLaMA Base [37]: We select ChatGLM2-6B (Base), ChatGLM3-6B (Base), and ChatGLM3-32B (Base) as the base models. In addition, we consider LLaMA-2-7B and LLaMA-2-13B as baselines.
- Continue Pre-training: We include Galactica [1] although we note that its continued pre-training does not necessarily include college-level scientific reasoning.
- Dataset-specific Tuning: For the dataset-specific tuning, we consider WizardMath [43], MAmmoTH-CoT [10], and MetaMath [11], which adapt to the MATH dataset.

## A.9   Detailed Experimental Results

Regarding domain-wise results in Figure 13, we present the detailed subject results in Table 9 and Table 10.

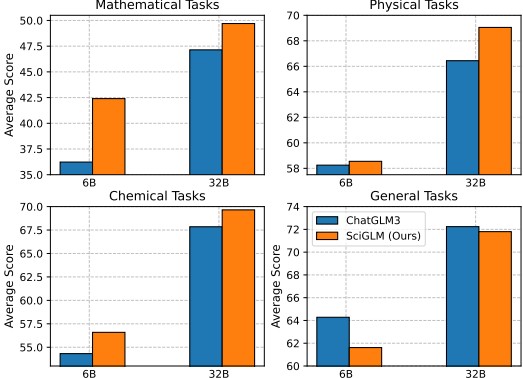

Figure 13: `SciGLM` consistently improving the ChatGLM3-Base for different scientific tasks, without sacrificing general language understanding capabilities.

Table 9: Detailed results of physical subjects for Figure 3. **Bold** denotes the best result in the closed-source models and open-source models.

| model | Ave. | CEval | | | | MMLU | | | | SciEval |
|---|---|---|---|---|---|---|---|---|---|---|
| | | *Avg.* | College Physics | High School Physics | Middle School Physics | *Avg.* | College Physics | Conceptual Physics | Middle School Physics | *Physics.* |
| ChatGLM3-6B-Base | 54.32 | *63.16* | 42.11 | 63.16 | 84.21 | *45.59* | **39.22** | 55.32 | 34.44 | 46.95 |
| SciGLM (ChatGLM3-6B-Base) | **56.59** | **66.67** | **47.37** | 63.16 | **89.47** | **46.52** | 34.31 | 55.32 | **41.06** | **47.56** |
| ChatGLM3-32B-Base | 67.85 | *78.95* | 57.89 | 78.95 | 100.00 | *56.76* | **42.16** | 62.55 | 57.62 | 51.22 |
| SciGLM (ChatGLM3-32B-Base) | **69.65** | **80.70** | **63.16** | 78.95 | 100.00 | **58.61** | 41.18 | **66.81** | 57.62 | **56.71** |

Table 10: Detailed results of chemical subjects for Figure 3. **Bold** denotes the best result in the closed-source models and open-source models.

| model | Ave. | CEval | | | | MMLU | | | SciEval |
| | | *Avg.* | College Chemistry | High School Chemistry | Middle School Chemistry | *Avg.* | College Chemistry | High School Chemistry | *Chemistry.* |
|---|---|---|---|---|---|---|---|---|---|
| ChatGLM3-6B-Base | 58.25 | *66.67* | 45.83 | 63.16 | 95.00 | *49.83* | 45.00 | **52.22** | 55.76 |
| SciGLM (ChatGLM3-6B-Base) | **58.55** | *68.25* | **41.67** | **73.68** | **95.00** | *48.84* | **47.00** | 49.75 | **57.27** |
| ChatGLM3-32B-Base | 66.44 | *69.84* | 41.67 | 78.95 | 95.00 | *63.04* | 51.00 | 68.97 | 60.15 |
| SciGLM (ChatGLM3-32B-Base) | **69.05** | *71.42* | **45.83** | 78.95 | 95.00 | *66.67* | 57.00 | 71.43 | **61.40** |

## A.10 Output Examples of `SciGLM`

We show several concrete examples in SciBench, as well as the output generated by `SciGLM` (32B) and ChatGLM3-32B-Base in Figure 14, Figure 15, Figure 16, Figure 17, and Figure 18.

---

**Problem** When an electron in a certain excited energy level in a one-dimensional box of length 2.00 \u00c5 makes a transition to the ground state, a photon of wavelength 8.79 nm is emitted. Find the quantum number of the initial state.

---

**Correct Answer**: 4

---

**Predicted Solution by ChatGLM3-32B-Base** (Served as baseline)

The ground state has $n = 1$. The energy in the $n = 1$ state is $\frac{1}{L^2}$. For the $n = 2$ state, this becomes $\left(\frac{1}{2L}\right)^2$.

Thus, this energy difference is $\frac{1}{2^2 2} - \frac{1}{1^2} = \frac{9}{4}$.

By de Broglie's equation, $hc/\lambda = \frac{9}{4hc\nu}$, so $\nu = \frac{4hc}{9\lambda}$. Thus, $E = h\nu = hc\frac{4}{9\lambda}$.

Therefore, $n = 2$, thus the answer is $\boxed{2}$. ✗

---

**Predicted Solution by `SciGLM`** (Finetuned from ChatGLM3-32B-Base)

Concept: The concept of this problem is based on the photoelectric effect. The photoelectric effect is the emission of electrons when light shines on a material. The photoelectric effect occurs when photons with energies greater than the binding energy of electrons cause the ejection of electrons from a material. When a photon is emitted by an electron, the energy of the photon is equal to the energy of the electron in the excited state minus the energy in the ground state. The energy of a photon is given by the equation E = hc/λ, where h is Planck's constant ($6.626 \times 10^{-34}$ Js), c is the speed of light ($3.000 \times 10^8$ m/s), and λ is the wavelength of the photon in meters. The energy of an electron in a one-dimensional box is given by the equation $E_n = (n^2 \times h^2/8l^2)$, where n is the quantum number of the state and l is the length of the box in meters. The problem asks for the quantum number of the initial state. This can be found by setting the energies of the initial and final states equal to the energy of the photon.

Step 1: Convert the length of the box and the wavelength of the photon to meters.
- The length of the box is given as 2.00 Å. We need to convert this to meters by multiplying by $10^{-10}$ (since 1 Å = $10^{-10} m$).
- The wavelength of the photon is given as 8.79 nm. We need to convert this to meters by multiplying by $10^{-9}$ (since 1 nm = $10^{-9}$ m).

Step 2: Calculate the energy of the photon.
- Use the equation E = hc/λ, where h is Planck's constant ($6.626 \times 10^{-34}$ Js) and c is the speed of light ($3.00 \times 10^8$ m/s).

Step 3: Set the energy of the final state (n = 1) equal to the energy of the photon and solve for n.
- Use the equation $E_n = (n^2 * h^2/8l^2)$, where l is the length of the box in meters.

Step 4: Convert the answer to scientific notation and round to 3 decimal places.

To sum up, the answer to the question is therefore $\boxed{n = 4.000}$. ✓

---

Figure 14: An example **Quantum Chemistry** problem in SciBench that has been solved accurately with `SciGLM` (32B). It shows that after instruction-tuning, the `SciGLM` learns skills/behaviors to first analyze the knowledge required for each problem, and then step-by-step solve the problem with correct formula and calculations.

## A.11 Examples of `SciInstruct`

We present `SciInstruct` examples for Math (Figure 19), Physics (Figure 20), Chemistry (Figure 21), and Formal Proofs (Figure 22).

**Problem** Extend Example 1.4-6 to an $n$-sided die. That is, suppose that a fair $n$-sided die is rolled $n$ independent times. A match occurs if side $i$ is observed on the $i$ th trial, $i = 1, 2, \ldots, n$. Find the limit of this probability as $n$ increases without bound.

**Correct Answer**: $1 - 1/e = 0.6321\ldots$

**Predicted Solution by ChatGLM3-32B-Base**
We need the limit of $P(A)$ as $n$ goes to infinity. $P(A)$ equals $\frac{1}{n^n}$ as there are $n$ possibilities for each of the $n$ rolls. Thus we wish to find $\lim_{n \to \infty} \frac{1}{n^n} = \lim_{x \to \infty} \frac{1}{x^x} = 0$.
Conclusion: The answer is therefore $\boxed{0}$. ✗

**Predicted Solution by SciGLM** (Finetuned from ChatGLM3-32B-Base)
The probability that all $i$ sides of the die are rolled $i$ times in a row is $\left(\frac{1}{n}\right)^n = \frac{1}{n^n}$.
Step 1: If we write the sum of the $n$ terms of this geometric sequence from $i = 1$ to $n$, it equals 1 because each term in the sum is $\frac{1}{n}$ and there is a total of $n$ terms being summed. Thus, if we want to find the probability that no match occurs, we sum all probabilities that no match occurs over $k = 1$ to $n$. This becomes

$\sum_{k=1}^{n} \frac{1}{k^n} = \frac{1}{1^n} + \frac{1}{2^n} + \frac{1}{3^n} + \cdots + \frac{1}{n^n}$

As $n \to \infty$, the sum approaches a limiting term $e^{-1}$. Since the total probability must equal 1 for all $n$, and we just established that the limit of the sum of all probabilities that no match occurs is $e^{-1}$, this must be the probability that no match occurs at all. The probability that at least one match occurs is then $1 - e^{-1}$.

Step 2:
Thus, as $n \to \infty$, the limit of the probability that no match occurs is $e^{-1} = 0.367$, and the limit of the probability that at least one match occurs is $1 - e^{-1} \approx 0.63$.

To sum up, the answer to the question is 0.63.

The limit of the probability that at least one match occurs is $\boxed{0.63}$. ✓

Figure 15: An example statistics problem in SciBench that is accurately solved with the `SciGLM` (32B).

**Problem** The World Series in baseball continues until either the American League team or the National League team wins four games. How many different orders are possible (e.g., ANNAAA means the American League team wins in six games) if the series goes four games?

**Correct Answer Latex**: 2
**Correct Answer Number**: 2

**Answer generated by ChatGLM3-32B-Base**
The answer is therefore $\boxed{20}$. ✗

**Answer generated by SciGLM (32B)**
Solution:
Step 1: American league team wins in 4 games $\binom{4}{4}$.
National team wins in 4 games $\binom{4}{4}$.

Step 2: Total number of ways
$= \binom{4}{4} + \binom{4}{4} = 2 \times 1 = 2$.

Conclusion: The answer is therefore $\boxed{2}$. ✓

Figure 16: A statistics example problem in SciBench that has been solved accurately with `SciGLM` (32B).

**Problem** 7.3-9. Consider the following two groups of women: Group 1 consists of women who spend less than 500 annually on clothes; Group 2 comprises women who spend over 1000 annually on clothes. Let $p_1$ and $p_2$ equal the proportions of women in these two groups, respectively, who believe that clothes are too expensive. If 1009 out of a random sample of 1230 women from group 1 and 207 out of a random sample 340 from group 2 believe that clothes are too expensive,
(a) Give a point estimate of $p_1 - p_2$.

**Correct Answer Latex**: $0.2115$
**Correct Answer Number**: 0.2115

**Answer generated by ChatGLM3-32B-Base**
To find the point estimate of $p_1 p_2$, which is the joint probability that a woman from group 1 and a woman from group2, believe that clothes are too expensive, we will use the number of women in each group that believe clothes are too expensive.

From group1, we have that 10.0907 of women believe that clothes are too expensive, And from group 2, we have that 0.6076 of women believe that clothes are too expensive.
The point estimate is approximately(123)=0.000000.
The answer is therefore 0.000 . ✗

**Answer generated by SciGLM (32B)**
Solution:
$n_1 = 1230, p_1 = \frac{1009}{n_1} \approx 0.819,$
$n_2 = 340, p_2 \approx \frac{207}{340} = 0.609$
$p_1 - p_2 \approx 0.819 - 0.609 = 0.210$
The answer is therefore 0.210 . ✓

Figure 17: A statistics example problem in SciBench that has been solved accurately with SciGLM (32B).

**Problem** Calculate the uncertainty $\Delta L_z$ for the hydrogen-atom stationary state: $2p_z$.

**Correct Answer Latex**: 0
**Correct Answer Number**: 0

**Answer generated by ChatGLM3-32B-Base**
Since the angular momentum of a revolving electron around the nucleus is: $L_z = \mu_B \hbar$. And the radius of revolution for the electron around the proton is about . The uncertainty is: $\Delta L_z = \frac{\hbar}{2}\sqrt{2}$. The answer is therefore 7.605 . ✗

**Answer generated by SciGLM (32B)**
Solution:
$\Delta L_z = \frac{h}{2\pi}\sqrt{\frac{\hbar}{mc\hbar c}}\sqrt{\frac{n(n-1)}{2}}$
$= \sqrt{\frac{1 \cdot (1-1)}{2}}$
$= \sqrt{\frac{0}{2}} = 0$
The answer is therefore 0 . ✓

Figure 18: A Quantum Chemistry example problem in SciBench that has been solved accurately with SciGLM (32B).

**Problem**
Using the method of undetermined coefficients to answer the below question.
$y'' - 2y' + 5y = \sin x, y(0) = 1, y'(0) = 1$

**Answer**
**Analysis:** The key points of this problem are the undetermined coefficients and the differential equation of parameter variations.
This test uses the Method of undetermined coefficients to solve differential equation problems.
Step1: the auxiliary equation is $r^2 - 2r + 5 = 0$ with roots $r = 1 \pm 2i$, so the complementary solution is:
$\iota_e(x) = e^x(c_1 \cos 2x + c_2 \sin 2x)$.
Try the particular solution:
$y_p(x) = A \cos x + B \sin x$,
$y'_p = -A \sin x + B \cos x$

Step2: substituting, we have:
$(-A \cos x - B \sin x) - 2(-A \sin x + B \cos x) + 5(A \cos x + B(\sin x) = \sin x$
$\Rightarrow (4A - 2B) \cos x + (2A + 4B) \sin x = \sin x$.
Then:
$4A - 2B = 0, 2A + 4B = 1 \quad \Rightarrow \quad A = \frac{1}{10}, B = \frac{1}{3}$
and the general solution is:
$y(x) = y_1(x) + y_p(x) = e'(c_1 \cos 2x + c_2 2x) + \frac{1}{10} \cos x + \frac{1}{3} \sin x$
But, $1 = y(0) = c_1 + \frac{1}{10} \quad \Rightarrow \quad c_1 = \frac{3}{10}$
and $1 = y'(0) = 2c_2 + c_1 + \frac{1}{5} \quad \Rightarrow \quad c_2 = -\frac{1}{26}$

Step3: Thus the solution to the initial-value problem is:
$y(x) = e^x \left( \frac{9}{10} \cos 2x - \frac{1}{20} \sin 2x \right) + \frac{1}{10} \cos x + \frac{1}{5} \sin x$.

To sum up, the answer is:
$y(x) = e^x \left( \frac{9}{10} \cos 2x - \frac{1}{20} \sin 2x \right) + \frac{1}{10} \cos x + \frac{1}{5} \sin x$.

Figure 19: An instruction example on the math subject.

**Problem**

Two lightbulbs, one rated 30 W at 120 V and another rated 40 W at 120 V, are arranged in two different circuits.

a. The two bulbs are first connected in parallel to a 120 V source.

i. Determine the resistance of the bulb rated 30 W and the current in it when it is connected in this circuit.

ii. Determine the resistance of the bulb rated 40 W and the current in it when it is connected to this circuit.

b. The bulbs are now connected in series with each other and have a 120 V source.

i. Determine the resistance of the bulb rated 30 W and the current in it when it is connected in this circuit.

ii. Determine the resistance of the bulb rated 40 W and the current in it when it is connected to this circuit.

c. In the spaces below, number the bulbs in each situation described, in order of their brightness.

(1= brightest, 4 = dimmest)

\_\_\_\_30 W bulb in the parallel circuit

\_\_\_\_40 W bulb in the parallel circuit

\_\_\_\_30 W bulb in the series circuit

\_\_\_\_40 W bulb in the series circuit

d. Calculate the total power dissipated by the two bulbs in each of the following cases.

i. The parallel circuit

ii. The series circuit

**Answer**

**Aanalysis:** Use formulas to calculate resistance, current, and total power.

(a) i: Calculate the resistance and current of the light bulb.

$P = V^2/R$ given R=480 $\Omega$ and

V=IR given I=0.25 A

ii: Calculate the resistance and current of the light bulb.

$P = V^2/R$ given R=360 $\Omega$ and

V=IR given I=0.33 A

(b) i./ii. Calculate the resistance and current of the light bulb.

The resistances are unchanged 480 $\Omega$ and 360 $\Omega$. The total resistance in series is 480 $\Omega$ + 360 $\Omega$ = 840 $\Omega$ making the total current $I = V/R = 0.14$ A which is the same value for both resistors in series.

(c) Compare the brightness of the light bulb.

The bulbs are brightest in parallel, where they provide their labeled values of 40 W and 30 W. In series, it is the larger resistor (the 30 W bulb) that glows brighter with a larger potential difference across it in series. This gives the order from top to bottom as 2 1 3 4.

(d) i: Calculate the total power consumed by two light bulbs.

In parallel, they each operate at their rated voltage so they each provide their rated power and$P_T = 30W + 40W = 70W$

ii: Calculate the total power consumed by two light bulbs

In series $\mathbf{P_T} = \mathbf{V_T}^2/\mathbf{R_T} = 17\mathbf{W}$

In summary,

(a) i: P = 480 $\Omega$ and V = 0.25A

ii: P = 360 $\Omega$ and V = 0.33A

(b) i/ii: P = 840 $\Omega$ and V = 0.14A

(c) This gives the order from top to bottom as 2 1 3 4.

(d) i: $P_T = 70W$

ii: $\mathbf{P_T} = 17\mathbf{W}$

Figure 20: An instruction example on the physics subject.

**Problem**

Consider a mixture of the two solids, $BaCl_2 + 2H_2O$ (FM 244.26) and KCl (FM 74.551), in an unknown ratio. (The notation $BaCl_2 \cdot 2H_2O$ means that a crystal is formed with two water molecules for each $BaCl_2$.) When the unknown is heated to $160°C$ for 1 h, the water of crystallization is driven off:

$$BaCl_2 \cdot 2H_2O(s) \xrightarrow{160°C} BaCl_2(s) + 2H_2O(g)$$

A sample originally weighing 1.7839 g weighed 1.5623 g after heating. Calculate the weight percent of Ba, K, and Cl in the original sample.

**Answer**

**Analysis:** The content of this question is to calculate the weight percentage.

Step1: Formula and atomic masses: $Ba(137.327), Cl(35.453), K(39.098), H_2O(18.015), KCl(74.551), BaCl_2 \cdot 2H_2O(244.26), H_2O$ lost $= 1.7839 - 1.5623 = 0.2216$ g $= 1.2301 \times 10^{-2}$ mol of $H_2O$. For $2\,molH_2O$ lost, 1 mol $BaCl_2 \cdot 2H_2O$ must have been present. $\frac{1}{2}\left(1.2301 \times 10^{-2}\ molH_2O\ \text{lost}\right) = 6.1504 \times 10^{-3}\ molBaCl_2 \cdot 2H_2O = 1.5024$ g.

The Ba and Cl contents of the $BaCl_2 \cdot 2H_2O$ are

$Ba = \left(\frac{137.33}{244.26}\right)(1.5024\ \text{g}) = 0.84469$ g

$Cl = \left(\frac{2(35.453)}{244.26}\right)(1.5024\ \text{g}) = 0.43613$ g

Step2: Because the total sample weighs 1.783 g and contains 1.5024 g of $BaCl_2 \cdot 2H_2O$, the sample must contain $1.7839 - 1.5024 = 0.2815$ g of KCl, which contains

$K = \left(\frac{39.098}{74.551}\right)(0.281_5) = 0.14763$ g

$Cl = \left(\frac{35.453}{74.551}\right)(0.281_5) = 0.13387$ g

Weight percent of each element:

$Ba = \frac{0.84469}{1.7839} = 47.35\%$

$K = \frac{0.14763}{1.7839} = 8.28\%$

$Cl = \frac{0.43613+0.13387}{1.7839} = 31.95\%$

In summary, the weight percent of Ba is 47.35%, the weight percent of K is 8.28%, the weight percent of Cl is 31.95%.

Figure 21: An instruction example on the chemistry subject.

**Problem**

lemma unfold_sub $\alpha$ [subtraction_monoid $\alpha$] (a b c : $\alpha$) (h : a + -b = c) : a - b = c :=.

**Answer**

by rw [sub_eq_add_neg, h]

**Problem** lemma subtype.exists_pi_extension $\iota$ : Sort* $\alpha$ : $\iota \to$ Sort* [ne : $\Pi$ i, nonempty ($\alpha$ i)]

p : $\iota \to$ Prop (f : $\Pi$ i : subtype p, $\alpha$ i) :

g : $\Pi$ i : $\iota$ , $\alpha$ i, ($\lambda$ i : subtype p, g i) = f :=

**Answer**

begin

tactic.classical,

refine $\langle \lambda$ i, if hi : p i then f $\langle$i, hi$\rangle$ else classical.choice (ne i), funext _$\rangle$,

rintro $\langle$i, hi$\rangle$,

exact dif_pos hi

end

Figure 22: Examples on formal proofs.

## A.12 Scientific Reasoning Datasets

**Mathematical Datasets.** In the past, the emphasis in mathematical reasoning tasks was primarily on fundamental arithmetic reasoning datasets [44; 45; 46], addressing basic mathematical problems such as AddSub [47]. Later on, to address realistic math word problems, some more difficult datasets are proposed [48; 49; 30; 50]. To construct a more difficult and diversified dataset, NumGLUE [51] and LiLA [52] have been introduced to enhance current research. However, their focus remains primarily on elementary, middle school, and high school math problems. Given the rapid advancement of LLMs, to assess their capacity and constraints in addressing more intricate math problems, MMLU [29] with college-level math problems has been proposed. To address the more challenging college-level math problems, PoT [53; 7] is proposed. Our proposed `SciInstruct` covers more complex and more diversified math problems at the college level.

**Scientific Datasets.** To address the scientific ability of LLMs, SciBench [7] proposes various solutions like calling existing tools (python and Wolfram). SciEval [6] presents a multi-level evaluation benchmark for scientific research. GPQA [8] builds a graduate-level Q&A benchmark to evaluate more difficult physics and chemistry questions.

## A.13 General Reasoning with LLMs

Assisted by Chain-of-Thought prompting [54; 55; 56] and Tree-or-Thought [57], LLMs have brought decent reasoning performance improvements. Especially, on the challenging BIG-Bench tasks, the CoT technique has already outperformed human ability [58]. Later on, to address reasoning tasks, a few studies [59; 60; 61; 62; 63; 11; 64; 65; 66] propose various methods to leverage LLMs and retain step-by-step processes. With the rapid development of LLM Agents, several works [67; 17] propose to utilize external tools to enhance the reasoning capabilities of LLMs. For example, ReAct [67] attempts to call existing tools like web search engines to improve the reasoning skills of LLMs. Indeed, leveraging the programs as thought processes is a natural way. Several recent studies have utilized programs as thought processes, such as PoT [68], to improve the reasoning abilities of LLMs. To enhance LLMs' reasoning performance in solving mathematical or scientific problems with PoT, several methods have been suggested, including Self-critic [69], Self-eval [70], and Plan-and-solve [71]. For example, self-critic [69] and self-eval [70] propose to adopt self-evaluation to improve the generated program's robustness. Different from these two methods, plan-and-solve [71] adopts more detailed planning instructions to generate a high-level reasoning plan for LLMs. Indeed, these methods have demonstrated that they can obtain great capabilities over PoT.

## A.14 Instruction Tuning in LLMs

To align language models with human preferences and effective objectives, some works [17; 72; 66] design instruction tuning. The instruction tuning aims to mine the potential capabilities by aligning with and responding to human preference. Early on, instruction tuning focuses on improving the instruction-following abilities of LLMs for general purposes. Represented by FLAN [73] and T0 [74], they aim to understand the generalization abilities of LLMs for instruction tuning. Later, to comprehend the efficacy and performance of enlarging the instructional tuning datasets on models, FLAN-v2 [75; 76] has been proposed to validate this goal. However, these methods build training instruction tuning datasets in a human-annotated manner. Recently, various studies [77; 72; 78; 17; 39; 79] have started to construct synthetic instruction following datasets distilled from some LLMs like GPT-3.5-turbo and GPT-4. Like these works, Platypus [80] constructs a small-scale instruction for the following dataset by utilizing a domain-specialized dataset, aiming to enhance the reasoning capabilities of LLMs.

# Part II

# Supplementary Materials

## B   Datasheet

### B.1   Motivation

**1. For what purpose was the dataset created?** *Our benchmark dataset was created to address the data scarcity challenge in the science domain and provide a suite of scientific instructions for training scientific language models capable of college-level scientific reasoning.*

**2. Who created the dataset and on behalf of which entity?** *The dataset was developed by LLM researchers (undergraduate students, doctoral students, and postdocs) listed in the author list.*

**3. Who funded the creation of the dataset?** *This work is supported by the NSFC 62276148, NSFC for Distinguished Young Scholar 62425601, a research fund from Zhipu, New Cornerstone Science Foundation through the XPLORER PRIZE and Tsinghua University (Department of Computer Science and Technology) - Siemens Ltd., China Joint Research Center for Industrial Intelligence and Internet of Things (JCIIOT).*

### B.2   Distribution

**1. Will the dataset be distributed to third parties outside of the entity (e.g., company, institution, organization) on behalf of which the dataset was created?** *Yes, the dataset is open to the public.*

**2. How will the dataset will be distributed (e.g., tarball on website, API, GitHub)?** *The dataset has been distributed through Hugging Face, Google Drive, Tsinghua Cloud, and the code used for developing baseline models through GitHub.*

**3. Have any third parties imposed IP-based or other restrictions on the data associated with the instances?** *No.*

**4. Do any export controls or other regulatory restrictions apply to the dataset or to individual instances?** *No.*

**5. When will the dataset be distributed?** *It has been released now.*

**6. Will the dataset be distributed under a copyright or other intellectual property (IP) license, and/or under applicable terms of use (ToU)?** *The dataset will be distributed under the CC BY 4.0 license.*

### B.3   Maintenance

**1. Who will be supporting/hosting/maintaining the dataset?** *Zhipu AI and THUDM will support, host, and maintain the dataset.*

**2. How can the owner/curator/manager of the dataset be contacted (e.g., email address)?** *The owner/curator/manager(s) of the dataset can be contacted through the following emails: Dan Zhang (zd18@tsinghua.org.cn) and Sining Zhoubian (zbsn21@mails.tsinghua.edu.cn).*

**3. Is there an erratum?** *No. If errors are found in the future, we will release errata on the main web page for the dataset (*https://github.com/THUDM/SciGLM *and* https://huggingface.co/datasets/zd21/SciInstruct*).*

**4. Will the dataset be updated (e.g., to correct labeling errors, add new instances, delete instances)?** *Yes, the datasets will be updated whenever necessary to ensure accuracy, and announcements will be made accordingly. These updates will be posted on the main web page for the dataset (*https://github.com/THUDM/SciGLM *and* https://huggingface.co/datasets/zd21/SciInstruct*).*

**5. If the dataset relates to people, are there applicable limits on the retention of the data associated with the instances (e.g., were the individuals in question told that their data would be retained for a fixed period of time and then deleted?)** *N/A*

**6. Will older version of the dataset continue to be supported/hosted/maintained?** *Yes, older versions of the dataset will continue to be maintained and hosted.*

**7. If others want to extend/augment/build on/contribute to the dataset, is there a mechanisms for them to do so?** *If others want to extend/augment/build on/contribute to the dataset, the most efficient way to reach us is via GitHub pull requests. For more questions, don't hesitate to get in touch with Dan Zhang (zd18@tsinghua.org.cn), who will be responsible for maintenance.*

## B.4 Composition

**1. What do the instance that comprise the dataset represent (e.g., documents, photos, people, countries?)** *Each instance includes a question, corresponding answer, and subject. These attributes are used to fine-tune LLMs capable of college-level scientific reasoning.*

**2. How many instances are there in total (of each type, if appropriate)?** *The constructed SciInstruct dataset comprises 254,051 instructions, including 123,869 physics and chemistry, 89,934 math, and 40,248 formal proof (Lean).*

**3. Does the dataset contain all possible instances or is it a sample of instances from a larger set?** *The datasets contain all possible instances.*

**4. Is there a label or target associated with each instance?** *Yes, each instance includes the corresponding subject.*

**5. Is any information missing from individual instances?** *No.*

**6. Are there recommended data splits (e.g., training, development/validation, testing)?** *We use all instances as a training set because there are various public benchmark datasets as test sets that cover a broad range of subjects.*

**7. Are there any errors, sources of noise, or redundancies in the dataset?** *No.*

**8. Is the dataset self-contained, or does it link to or otherwise rely on external resources (e.g., websites, tweets, other datasets)?** *The dataset is self-contained.*

**9. Does the dataset contain data that might be considered confidential?** *No.*

**10. Does the dataset contain data that, if viewed directly, might be offensive, insulting, threatening, or might otherwise cause anxiety?** *No.*

## B.5 Collection Process

**1. How was the data associated with each instance acquired?** *The data associated with each instance is acquired from a variety of sources, including textbooks, pedagogical materials, and problem sets. References for these sources are provided in Table 7 in Appendix 7.2.*

**2. What mechanisms or procedures were used to collect the data (e.g., hardware apparatus or sensor, manual human curation, software program, software API)?** *We first use Optical character recognition (OCR) to transform the question and final answer in the collected materials, then utilize a powerful LLM API to generate the intermediate reasoning, and finally filter out the traces with a correct predicted answer.*

**3. Who was involved in the data collection process (e.g., students, crowdworkers, contractors) and how were they compensated (e.g., how much were crowdworkers paid)?** *Regular LLM researchers (e.g., undergraduate students, PhD students, and postdocs listed in the author list) at Tsinghua and Caltech were involved in the data collection process.*

**4. Does the dataset relate to people?** *No.*

**5. Did you collect the data from the individuals in questions directly, or obtain it via third parties or other sources (e.g., websites)?** *We obtained the dataset from textbooks, pedagogical materials, and problem sets. References for these sources are provided in Table 7 in Appendix 7.2.*

### B.6    Uses

**1. Has the dataset been used for any tasks already?** *Yes, this dataset has been used to generate new datasets, for example, to convert the text-formatted calculations into code formats and to fine-tune the code-based language models.*

**2. What (other) tasks could be the dataset be used for?** *No.*

**3. Is there anything about the composition of the dataset or the way it was collected and preprocessed/cleaned/labeled that might impact future uses?** *The current composition of the datasets is self-sufficient to train a scientific language model. Any changes in the next release and updates will be documented and shared through the dataset webpage (*`https://github.com/THUDM/SciGLM`* and *`https://huggingface.co/datasets/zd21/SciInstruct`*).*

**4. Are there tasks for which the dataset should not be used?** *No.*

