# OpenReview forum: "SciInstruct: a Self-Reflective Instruction Annotated Dataset for Training Scientific Language Models"
_NeurIPS.cc/2024/Datasets_and_Benchmarks_Track — NeurIPS 2024 Track Datasets and Benchmarks Poster_

### Official Review · Reviewer_Wv6V · 2024-07-22

**Rating:** 6
**Confidence:** 4
**Correctness:** Yes
**Clarity:** Yes

**Review:**

Pros.:
1.	This paper provides a broad and diverse dataset encompassing physics, chemistry, math, and formal proofs, which addresses the scarcity of high-quality scientific data for training LLMs.
2.	 The self-reflective annotation framework allows LLMs to autonomously generate and refine step-by-step reasoning.
3.	The paper includes thorough analysis and benchmarking, demonstrating the effectiveness of SciInstruct in improving LLM capabilities across multiple scientific domains.
4.	The dataset and code are made publicly available, promoting transparency and enabling further research.

Cons.:
1. The scope of this paper is somewhat over-claimed. The authors mentioned that the dataset is used for training scientific language models while most benchmarks such as C-Eval, and MMLU are specifically the college-level scientific question-answering tasks. Authors are encouraged to clarify the scope of this work in the title and introduction.
2. The novelty of the self-reflective annotation framework is somehow limited. The ideas of CoT[1]  and Reflection[2] for improving LLM instruction-tuning are already discussed in some previous work. Authors are encouraged to discuss these related works and emphasize the key contribution of this work.
[1] Seungone Kim, et al. The CoT Collection: Improving Zero-shot and Few-shot Learning of Language Models via Chain-of-Thought Fine-Tuning. https://arxiv.org/abs/2305.14045
[2] Ming Li, et al. Reflection-Tuning: Data Recycling Improves LLM Instruction-Tuning. https://arxiv.org/abs/2310.11716

3. In line 170, the authors propose the challenge “LLM generates a correct answer, the intermediate reasoning can still be wrong”. To solve this challenge, authors train a classifier, but the training data for this classifier is filtered by “inaccurate answers ” and doesn’t actually contain the “accurate answers but wrong intermediate reasoning”. Hence, can this classifier solve the proposed challenge? It would be better to rethink the proposed challenge or conduct more experimental analysis about this. Additionally, for the classifier, considering scaling, if the size of the dataset increases, whether it needs more datasets to train the instruction-quality classifier?
4. One largest concerns about this work is the safety of the proposed dataset. Whether the dataset contain sensitive information and how to filter such information? For example, for chemistry, whether the datasets contain some methods to synthesize some toxic or dangerous chemicals? Authors are encouraged to conduct analysis about safety about the proposed datasets.
5. The performance of closed-source models such as GPT-4, and Claude in MMLU-Sci is not reported. Hence the Avg. Sci metrics are not reported and It’s difficult to evaluate the gap between SciGLM and closed-source models. From other benchmarks, the overall performance of SciGLM is worse than the closed-source models.

**Strengths:**

Please see Pros.

**Additional Feedback:**

No

**Documentation:**

Yes

**Ethics:**

Yes. Especially for the lack of safety analysis in the chemistry field.

**Limitations:**

Not, Please see Cons. especially for the lack of safety analysis of this work.

**Opportunities For Improvement:**

Please see Cons. especially for the lack of safety analysis of this work.

**Relation To Prior Work:**

Not very clear. Authors are encouraged to discuss:
1.	What are the differences between the proposed data annotation framework and the previous instruction dataset construction methods?
2.	In Table 1, the Authors mentioned that Galactica doesn’t contain Lean and the authors only identify four domains. Considering science, how about the biology, medical, etc part? Authors are encouraged to show a more comprehensive comparison to previous LLMs.

**Summary And Contributions:**

This paper introduces SciInstruct and a novel self-reflective instruction annotation framework to address the data scarcity challenge in the science domain. The contributions are: 1. this paper constructs a comprehensive science dataset 2. Propose an annotation framework to generate and refine step-by-step reasoning for scientific questions 3. Conduct extensive experiments to show to high quality of the constructed dataset.

---

> ### Author Rebuttal · Authors · 2024-08-17
>
> Thank you for acknowledging our contribution to a broad and diverse finetuning dataset, an effective self-reflective annotation framework, comprehensive benchmarks, and available code and dataset. Thank you for raising valuable opportunities for improvement. We appreciate your dedicated time and effort in thoroughly assessing our work. We provide a detailed response to each of them to address your concerns and questions.
>
> ```
> Q1: Concerns about the clarification of the scope of this paper.
> ```
> Thank you for your valuable suggestion. Our primary aim is to develop a scientific language model tailored for college-level understanding. As part of our evaluation, we have evaluated various college-level question-answering benchmarks, including C-Eval and MMLU, focusing on subjects typically encountered at the college level.
>
> To enhance the clarity and precision of our work, we will refine the scope of our study in both the title and introduction. We appreciate your feedback and are committed to enhancing the clarity and accuracy of our study's focus.
>
> ```
> Q2 & R1: Concerns about the novelty of the self-reflective annotation framework.
> ```
> Thank you for raising this important point. Let's address your concerns by comparing our work with the related studies mentioned.
>
> **1. Comparisons with these related works.**
>
> When discussing the comparison with the CoT Collection [1] by Seungone Kim et al. and the Reflection-Tuning by Ming Li et al., it's essential to note the distinct approaches each work takes. While the CoT Collection focuses on enhancing intermediate rationales and reasoning steps using chain-of-thought methodology, our work extends this approach by enabling step-by-step reasoning across multiple languages. Unlike smaller language models that struggle with multilingual tasks due to fine-tuning limitations, our methodology aims to efficiently reason by generating intermediate steps post fine-tuning with CoT data from various languages.
>
> Regarding Reflection-Tuning [2], which shares similarities with our work in the pursuit of generating high-quality instructions through reflection, our methodology introduces a unique three-stage process involving CoT prompts, reflective prompts, and direct hints in prompts to enhance the reflective generation process.
>
> [1] Seungone Kim, et al. The CoT Collection: Improving Zero-shot and Few-shot Learning of Language Models via Chain-of-Thought Fine-Tuning. https://arxiv.org/abs/2305.14045
>
> [2] Ming Li, et al. Reflection-Tuning: Data Recycling Improves LLM Instruction-Tuning. https://arxiv.org/abs/2310.11716
>
> **2. Key contributions of our work:**
>
> (1) Our motivation is to curate a broad and diverse college-level dataset to train scientific language models. This dataset has been made public to promote transparency and enable further research.
>
> (2) By leveraging the effectiveness of chain-of-thought reasoning to enhance LLM performance, we promote the generation of high-quality intermediate reasoning steps through our proposed self-reflective annotation framework.
>
> (3) Through rigorous analysis and benchmarking across multiple scientific domains, we demonstrate the efficacy of SciIntruct in enhancing LLM capabilities.
>
> We appreciate your feedback and are committed to refining our approach to contribute meaningfully to the field. Thank you for your valuable insights.
>
> ```
> Q3: Concerns on the training set of the instruction-quality classifier.
> ```
> Thank you for your insightful questions.
>
> (1) In response to your first inquiry, we acknowledge the concern regarding the training data for the classifier. While the current filtering process includes inaccurate answers, it is crucial to consider including instances where the answer is correct, yet the intermediate reasoning is incorrect. This aspect indeed poses a significant question on the efficacy of the classifier in solving the proposed challenge. It may be beneficial to reconsider the filtering criteria for the classifier training data to encompass cases where accurate answers are accompanied by incorrect intermediate reasoning. This adjustment could potentially enhance the classifier's ability to tackle the identified challenge effectively. We will update this challenge in our paper.
>
> (2) Addressing the second point, when contemplating scaling considerations for the classifier with an expanding dataset, there are two primary approaches to consider. Firstly, scaling the instruction-quality classifier itself by augmenting its size to accommodate the increased dataset volume. Alternatively, expanding the dataset used for training the classifier could also be a viable strategy to ensure its robustness and effectiveness as the dataset size grows.
>
> By evaluating these strategies and potentially combining them, we aim to enhance the classifier's capacity to handle larger datasets while maintaining its proficiency in assessing instruction quality.
>
> ```
> Q4: Concerns regarding the safety of the proposed dataset.
> ```
> Thank you for your inquiries. In Section 2 of our paper, we elaborate on how we curate the dataset from various sources, with a primary focus on question-and-answer instructions. Consequently, the dataset is devoid of sensitive information.
>
> Specifically, within the chemistry subject, the instructions are structured around question-and-answer formats. Our primary objective is to generate intermediate reasoning steps based on the final answer for questions that are from existing textbooks and websites, rather than detailing the synthesis of toxic or hazardous chemicals.
>
> Moreover, two Ethics Reviewers have independently affirmed that this research centers on scientific instruction. They have explicitly stated that our work does not raise ethical issues, pose significant risks, or directly facilitate the synthesis of toxic or dangerous substances.
>
> We are committed to ensuring the safety and integrity of our research, and we appreciate your attention to this critical aspect of our work.

---

> > ### Author Rebuttal · Authors · 2024-08-17
> >
> > ```
> > Q5: Concerns on results on MMLU-Sci.
> > ```
> > Thank you for your inquiries.
> >
> > We have diligently attempted to access detailed performance results for MMLU on GPT-4 and Claude-v1.3 across various subjects. Regrettably, due to the unavailability of these specific results, we were unable to include the MMLU-Sci analysis in our paper.
> >
> > In the absence of direct performance data for these closed-source models in the context of MMLU-Sci, as shown in Figure 3, we focused on comparing the performances of SciGLM (6B) and Sci: Mistral (7B) against existing benchmarks where evaluation results for all models were readily available.
> >
> > We appreciate your feedback and understand the importance of comprehensive comparisons. Our decision to exclude MMLU-Sci results was based on the unavailability of specific performance metrics for the closed-source models mentioned. We remain committed to providing thorough and transparent evaluations in our research.
> >
> > Thank you for highlighting these aspects, we are dedicated to addressing any gaps in our analysis to enhance the overall quality of our findings.
> >
> > ```
> > R2: Concerns on results on Galactica.
> > ```
> > Thank you for your queries.
> >
> > In our study, we have provided comparisons of previous large language models (LLMs) such as Galactica-6.7B and Galactica-30B in Figure 3, Table 4, and Table 5. These comparisons span across diverse benchmarks encompassing subjects like Mathematics, Physics, and Chemistry.
> >
> > While our focus has been on these specific domains in our comparisons, we acknowledge the importance of a more comprehensive evaluation that includes areas like Biology and Medicine. Although Galactica does not incorporate Lean and other specific domains, our analysis aims to highlight the performance of these models in key subject areas to provide insights into their capabilities across a range of disciplines.
> >
> > We appreciate your feedback and understand the value of expanding our comparisons to offer a more holistic view of the performance of Galactica and other LLMs. Your input will guide us in enhancing the depth and breadth of our comparative analyses in future research. Thank you for your constructive insights.
> >
> > Once again, we sincerely thank you for your thoughtful evaluation and valuable suggestions, which have greatly contributed to improving our work. We believe that the revisions we have made adequately address your concerns and questions regarding the novelty and safety of SciInstruct, the results on MMLU-Sci and Galactica, and instruction classifies. If you believe that our responses have satisfactorily addressed your concerns about the issues, we kindly request that you consider adjusting the final evaluation to reflect this.

---

### Official Review · Reviewer_UPD9 · 2024-07-22
**Good instruction tuning dataset paper**

**Rating:** 7
**Confidence:** 4
**Correctness:** Claims made in this paper are correct…
**Clarity:** Paper is well-written.

**Review:**

This paper presents a high quality instruction tuning dataset that has been shown to outperform other datasets when used to finetuned popular pretrained LLMs. The evaluation performed is of high quality as well. The contribution in this work in terms of dataset and curation strategy is significant.

**Strengths:**

1. This paper provides an effective finetuning dataset for that improves downstream performance of several popular LLMs in a wide range of common evals.
2. The authors leverage the observation that chain-of-thought reasoning improves LLM performance, and created an instruction tuning dataset that encourages this behavior.
3. The authors develop an effective self-annotated pipeline for curating this data, and provides transparent information to recreate this.

**Additional Feedback:**

See "Opportunities For Improvement"

**Documentation:**

Good documentation is provided in terms of data curation, with code provided in github.

**Ethics:**

No ethical concerns spotted.

**Limitations:**

Limitations on model range and potential future directions are well discussed.

No error bars have been reported with the result, i.e., experiments have been run a single time. But given the breadth of the eval and cost of large models investigated here, this is understandable.

**Opportunities For Improvement:**

1. This reviewer is confused with table 4 and 5  -- why is the SciIntruct few shot, but vanilla measured with one shot? This doesn't seem like an apples to apples comparison to me.
2. What are the tokenizers used for the different models? This could fit in the appendix
3. SciInstruct has 244051 verified instructions -- can you also provide the number of tokens (as well as the tokenizer used for each model). This could fit in the appendix.

**Relation To Prior Work:**

Prior work is well discussed.

**Summary And Contributions:**

The authors have released a solid open dataset, while releasing a transparent curation strategy. This dataset is shown to improve the performance of various LLMs in a number of well-accepted evaluations. This is a significant contribution to the LLM research as a whole. I recommend this paper for acceptance with the following minor comments in "Opportunities for improvement":

---

> ### Author Rebuttal · Authors · 2024-08-17
>
> Thank you for acknowledging our contribution to an effective finetuning dataset, interesting observation, and an effective self-reflective annotation pipeline. Thank you for raising valuable opportunities for improvement. We appreciate your dedicated time and effort in thoroughly assessing our work. We provide a detailed response to each of them to address your concerns and questions.
>
> ```
> Q1: Concerns about the evaluation of few-shot or one-shot scenarios in Table 4 and Table 5.
> ```
> Thank you for raising this concern. In reference to the performance of vanilla models, we either utilize the publicly reported results or assess them under consistent experimental conditions, which encompass both zero-shot and few-shot scenarios.
>
> Concerning SciInstruct, we follow a similar approach, not restricting the evaluation zero-shot to the few-shot scenario, and subsequently report the best-performing results. This methodology is detailed in our paper from line 219 to line 221.
>
> Furthermore, the "+ SciInstruct" notation in both Table 4 and Table 5 signifies that we fine-tuned the backbone models (ChatGLM3-6B-Base/ChatGLM3-32B-Base, LLAMA3-8B-Instruct, and Mistral-7B: MetaMATH) using SciInstruct. This process is independent of the few-shot or zero-shot distinctions.
>
> We appreciate your attention to this aspect of our evaluation methodology and are committed to ensuring clarity in our reporting of results.
>
> ```
> Q2 & Q3: Concerns regarding the tokenizers and the number of tokens used for different LLMs.
> ```
> Thank you for your inquiries. We will address both questions together to provide a comprehensive understanding of the tokenizers used in our study. In our research, SciInstruct has 244,051 verified instructions. We have employed three Language Model Models (ChatGLM3, Llama3-8B-Instruct, and Mistral-7B: MetaMATH) as the backbone to assess the efficacy of our proposed datasets. Below, we present the tokenizers utilized for each model, along with their respective sources and the total number of tokens used:
>
> | Model | Tokenizer | Source | The number of tokens
>  --- | --- | --- | ---
> | ChatGLM3-6B-Base| Byte-level byte pair encoding (BPE) |  Section 2 in https://arxiv.org/pdf/2406.12793  | 242,346,686
> | ChatGLM3-32B-Base | Byte-level byte Pair Encoding (BPE) | Section 2 in https://arxiv.org/pdf/2406.12793  | 242,346,686
> | Llama3-8B-Instruct |Byte Pair Encoding (BPE) | Source on https://github.com/meta-llama/llama3/issues/60#issuecomment-2066877260  | 195,723,401
> | Mistral-7B: MetaMATH | Byte-fall back BPE |https://huggingface.co/mistralai/Mistral-7B-v0.1  | 242,728,050
>
>
> We see that three LLMs, ChatGLM, LLAMA, and Mistral, use the same type when it comes to Tokenizer.  Both use a tokenizer based on Byte Pair Encoding (BPE) to process text input data. Byte Pair Encoding is a popular word segmentation method that splits text into subwords or characters so that models can better understand and process text data.
>
> We appreciate your question and suggestion. In response to your feedback, we will include this detailed information in the Appendix section of our revised manuscript. Thank you for highlighting the importance of providing clarity on the tokenization process for each model.
>
> Once again, we sincerely thank you for your thoughtful evaluation and valuable suggestions, which have greatly contributed to improving our work. We believe that the revisions we have made adequately address your concerns and questions regarding the tokenizer and the number of tokens. If you believe that our responses have satisfactorily addressed your concerns about the issues, we kindly request that you consider adjusting the final evaluation to reflect this.

---

> > ### Comment · Reviewer_UPD9 · 2024-08-30
> >
> > I already recommended the paper for an accept. So no extra comments. Thank you for addressing my queries irregardless

---

### Official Review · Reviewer_kniG · 2024-07-24
**A Q/A dataset containing reasoning traces**

**Rating:** 6
**Confidence:** 4
**Correctness:** Yes
**Clarity:** Yes, most of the manuscript is well w…

**Review:**

It is novel and significant to supplement Q/A with step-by-step reasoning, which should make this dataset more useful. The manuscript is in general well written, and can be further improved.

**Strengths:**

* Include college-level scientific Q/As.
* Include reasoning traces of Q/A, annotated by the proposed process of self-reflective critic-and-revise
* The dataset is multi-lingual
* Both the dataset and benchmark are open-source and transparent.

**Additional Feedback:**

N/A

**Documentation:**

Yes

**Limitations:**

Yes

**Opportunities For Improvement:**

Some wordings are confusing. For example, in the Limitation section, “In line with what was discussed in Section 2.3, we employ an instruction-quality classifier to boost the instruction quality, yielding improved performance as shown in Table 2. However, we anticipate that the instruction-quality classifier, also referred to as the reward model, could provide even greater benefits.”

More evidence is needed to convince readers the claim "without sacrificing the language understanding capabilities of the base model". Or revise it to be more precise.

Table 5, label the best performance for each dataset.

**Relation To Prior Work:**

Yes.

**Summary And Contributions:**

Large Language Models (LLMs) struggle with complex scientific concepts, symbolic equations, and advanced scientific computing. A self-reflective instruction annotated dataset SciInstruct is introduced for fine-tuning LLMs to enhance their scientific and mathematical reasoning capabilities. SciInstruct has a broader coverage of scientific domains (physics, chemistry, math, and formal proofs) than existing dataset. In addition, the Q/A pairs in existing datasets, SciInstruct contains the reasoning traces of Q/A as instructions. The value of SciInstruct was demonstrated by conducting experiments using three LLMs.

---

> ### Author Rebuttal · Authors · 2024-08-17
>
> Thank you very much for acknowledging the strengths of this work as curating college-level scientific Q/As, the proposed self-reflective critic-and-revise framework, multi-lingual dataset, and open-source and transparent dataset and benchmark. Thank you for raising valuable opportunities for improvement. We appreciate your dedicated time and effort in thoroughly assessing our work. We provide a detailed response to each of them to address your concerns and questions.
>
> ```
> Q1: Concerns regarding the instruction-quality classifier or reward model in the Limitation section.
> ```
> Thank you for your question. We implemented an instruction-quality classifier to enhance the performance of our model, based on ChatGLM3-6B-Base at a smaller scale, as mentioned from line 181 to line 183. Notably, if we were to scale up the base model to sizes like 12 - 13B or 32 - 34B, the classification performance would be better. This rationale was outlined in the Limitation section. We will provide additional elaboration on this question in the Limitation section in our upcoming revision.
>
> ```
> Q2: Concerns about maintaining language understanding capabilities.
> ```
> We appreciate your feedback. To assess SciGLM's impact on general language understanding capabilities, we conducted evaluations on tasks such as MMLU, CEval, and MBPP, detailed in Section 3.2 and Table 6. As highlighted from line 236 to line 242, SciGLM demonstrates slight performance enhancements on standard tasks like MMLU and CEval. While performance on code generation tasks like MBPP is slightly lower, the overall consistency is maintained.
>
> We will refine this statement in the Abstract to enhance precision.
>
> ```
> Q3: Missing labels of the best performance for each dataset in Table 5.
> ```
> Thank you for your suggestions. We will ensure that the best performance for each dataset is clearly labeled in our manuscript. We appreciate your valuable feedback and are committed to incorporating these improvements in our next revision.
>
> Once again, we sincerely thank you for your thoughtful evaluation and valuable suggestions, which have greatly contributed to improving our work. We believe that the revisions we have made adequately address your concerns and questions regarding the descriptions of Limitations and language understanding abilities. If you believe that our responses have satisfactorily addressed your concerns about the issues, we kindly request that you consider adjusting the final evaluation to reflect this.

---

### Official Review · Reviewer_KU1m · 2024-07-25
**Review of SciInstruct: a Self-Reflective Instruction Annotated Dataset for Training Scientific Language Models**

**Rating:** 7
**Confidence:** 4
**Correctness:** Yes
**Clarity:** The writing style can be improved to …

**Review:**

The dataset size is substantial and curation process ensures quality reasoning steps. As demonstrated by the performance of finetuned models, the dataset will be useful to the community. While the paper does present significant details and statistics, the writing could be improved by reorganizing the sections.

**Strengths:**

The targeted domain of college-level scientific reasoning is interesting as it includes physics, chemistry, math, and formal theorem proofs in Lean.
The dataset size in the submission is substantial.
The finetuned models indicate that the dataset is useful in improving the reasoning capabilities of LLMs.

**Additional Feedback:**

NA

**Documentation:**

Yes, the dataset is available on huggingface.

**Limitations:**

Yes

**Opportunities For Improvement:**

Could the authors explain why there are only 42,497 instances to startwith in Figure 5 out of the original  257,143? If the rest are correctly inferred by GPT-4 in the Self-Reflective Instruction Annotation stage, then it means that 214646 (84% of the dataset) is easily answered by GPT4? Does that mean that those 84% questions are relatively simpler?
I could see some instances in chinese in the dataset. It might be informative to report the statistics in the paper.
I maybe missed it, but how are questions classified as simple solution vs complex solutoin as depicted in Figure 2?
A lot of tables and figures in the Appendix are referenced which interrupt the reading flow.
Is there any specific reason why the section/table/figure references are in red?

**Relation To Prior Work:**

It would be great if the related work can be included in the main paper instead of the appendix. A clear description of the differences with existing work such as ScienceQA, SciBench would be helpful.

**Summary And Contributions:**

The authors present SciInstruct, a scientific instruction dataset to enhance the college-level scientific reasoning of LLMs, spanning content from physics, chemistry, math, and formal theorem proofs in Lean, of size 257,143. The authors propose a self-reflective annotation framework to generate high-quality reasoning steps. Finally, the authors fine-tune different LLMs on the SciInstruct dataset to verify the effectiveness of the dataset. Results indicate improvements on various scientific and mathematical benchmarks, without sacrificing general language understanding tasks.

---

> ### Author Rebuttal · Authors · 2024-08-17
>
> Thank you for acknowledging our contribution to an interesting targeted domain, a substantial and useful dataset for the community, and a self-reflective annotation framework, and raising valuable opportunities for improvement. We appreciate your dedicated time and effort in thoroughly assessing our work. We provide a detailed response to each of them to address your concerns and questions.
>
> ```
> Q1: Concerns about the number of instances in Figure 5 (The workflow for self-reflective instruction annotation framework) and the number of original instructions.
> ```
> We appreciate the reviewer's attention to this concern. Allow us to clarify this issue:
>
> Firstly, in Table 3, SciInstruct encompasses subjects like Physics & Chemistry, Math, and Formal Proofs, presented in both Chinese and English (as indicated in line 105).
>
> For the Physics & Chemistry subject exemplified in Figure 5, the initial set comprised 123,869 instructions, with 42,497 in Chinese and 81,372 in English translated from Chinese. Therefore, Figure 5 presents the self-reflective instruction annotation stage for Physics & Chemistry subjects in Chinese.
>
> The workflow and filter applied to English instructions mirror those for Chinese instructions.
> Consequently, 46.65% of the dataset can be readily answered by GPT-4 in the initial stage.
>
> Thank you for your question again. We will revise the caption of Figure 5 accordingly.
>
> ```
> Q2: Concerns about the proportion of the Chinese dataset.
> ```
> Thanks for your question. Building upon the aforementioned details, the Chinese dataset predominantly consists of Physics & Chemistry, and Math.
>
> Specifically, Physics & Chemistry subjects comprised 52,952 Chinese instructions post Figure 5 workflow application. Math contained 14,934 Chinese instructions, totaling 67,886 instructions. Therefore, the Chinese dataset constitutes 26.72% of the overall dataset.
>
> We appreciate your inquiry and suggestion. We will report these statistics in this paper to provide information.
>
> ```
> Q3: Concerns regarding question classification as simple vs. complex solutions as depicted in Figure 2.
> ```
> We appreciate your inquiry. Typically, we classify questions based on the type of answer they require. For instance, single-choice and gap-filling questions are categorized as simple solutions, while short answer questions and those necessitating multiple steps are deemed complex solutions.
>
> Thank you for your question. We will include these clarifications in our forthcoming version.
>
> ```
> Q4: Concerns about red section/table/figure references.
> ```
> Thank you for highlighting this. The red color of the section/table/figure references is a default setting in LaTeX. In the upcoming version, we will standardize these references to black. We are grateful for your feedback and look forward to incorporating these improvements in the next iteration of our work.
>
> ```
> R1: Concerns about the location of related work.
> ```
> We value your suggestion and agree to relocate pertinent related work from the appendix to the main paper in our final version, which would be helpful to readers to understand the differences between this paper and the existing work.
>
> Once again, we sincerely thank you for your thoughtful evaluation and valuable suggestions, which have greatly contributed to improving our work. We believe that the revisions we have made adequately address your concerns and questions regarding the number of SciInstruct and the proportion of the Chinese dataset. If you believe that our responses have satisfactorily addressed your concerns about the issues, we kindly request that you consider adjusting the final evaluation to reflect this.

---

### Author Rebuttal · Authors · 2024-08-17

Dear ACs and Reviewers,

thank you very much for your valuable feedback. We list the main issues raised by reviewers and explain them below.

```
The motivation and key contributions of SciInstruct.
```
We provide detailed descriptions in response to Reviewer Wv6V's queries (Q2 and R1). Specifically,

* Our motivation is to curate a broad and diverse college-level dataset to train scientific language models. This dataset has been made public to promote transparency and enable further research.

* By leveraging the effectiveness of chain-of-thought reasoning to enhance LLM performance, we promote the generation of high-quality intermediate reasoning steps through our proposed self-reflective annotation framework.

* Through rigorous analysis and benchmarking across multiple scientific domains, we demonstrate the efficacy of SciInstruct in enhancing LLM capabilities.

```
The statistics and details of curated SciInstruct.
```
* We illustrate the total number of instructions in Figure 5 and elaborate on the proportion of the Chinese dataset in response to Reviewer KU1m's queries (Q1 and Q2).
* We describe the different question types in our dataset in response to Reviewer KU1m's query (Q3).
* We detail the tokenizer used and the respective number of tokens employed for different models in response to Reviewer UPD9's queries (Q2 and Q3).
* We address the safety concerns associated with our proposed dataset in response to Reviewer Wv6V's query (Q4).

```
 The design of the instruction-quality classifier.
```
* We delve into the description of the Limitation section in response to Reviewer kinG's question (Q1).
* We elaborate on the design of the training set of the classifier in response to Reviewer Wv6V's query (Q3).

```
The performance of curated SciInstruct.
```
* We discuss the model's impact on general language capabilities in response to Reviewer kinG's query (Q2).
* We clarify the evaluation setting, including few-shot and zero-shot scenarios, in response to Reviewer UPD9's question (Q1).

```
The results of well-trained LLMs.
```
* We provide an explanation of the Avg. Sci results in response to Reviewer Wv6V's query (Q5).
* We outline the comparison between our models and Galactica in response to Reviewer Wv6V's inquiry (R2).

```
Presentation of this paper.
```
* We commit to updating the color scheme as suggested in response to Reviewer KU1m's query (Q4).
* We will adjust the location of the Related Work section as per Reviewer KU1m's feedback (R1).
* We will include labels indicating the best performance in our response to Reviewer kinG's question (Q3).
* We will enhance the clarification of the paper's scope in response to Reviewer Wv6V's inquiry (Q1).

Your valuable feedback guides us in improving the clarity and quality of our work. We appreciate your insights and are dedicated to addressing all suggestions for a more comprehensive and polished presentation. Thank you for your valuable contribution.

---

### Comment · Area_Chair_ALnU · 2024-08-29
**Reminder to review comments before end of discussion period on 8/31**

Reviewers, thank you for your time and contributions thus far. This is a reminder that the discussion period ends in two days on August 31. Please take some time to engage with the authors' comments and adjust scores if appropriate.

---

### Decision · Program_Chairs · 2024-09-26

**Decision:**

Accept (Poster)

**Comment:**

This paper introduces SciInstruct, a dataset for scientific reasoning, along with a novel self-reflective instruction annotation framework, demonstrating improvements in fine-tuned LLM performance in scientific reasoning. This work would be interesting for the community. I believe major concerns have been addressed and trust the authors to make additional changes by the camera-ready deadline if accepted.

**Quality**: The dataset is well-curated, with an interesting annotation process using self-reflective critic-and-revise. Finetuning on the dataset improves LLM performance.

**Clarity**: The manuscript is generally well-written but suffers from organizational issues and some confusing sections. Dataset statistics and classification criteria require further clarification as per R1. The authors should also add any missing details on tokenizers and token counts as per R3’s feedback.

**Originality**: The paper's approach is original in its use of step-by-step reasoning and self-reflective annotation to create a scientific reasoning dataset.

**Significance**: The dataset and annotation framework offer valuable contributions to the field by enhancing scientific reasoning in LLMs. Some claims, such as maintaining base model language understanding without sacrifice, are overstatements and require reconsideration.